# Cell-type-specific immune programs orchestrate spatial defense in the Arabidopsis leaf epidermis

Jingpu Song [1], Mahsa Modareszadeh[1], Dinithi Kumarapeli[1], Wilson Andres Acosta[1], Yuhai Cui [2] & Yangdou Wei [1] ✉

Sessile plants rely on individual cells to perceive pathogens and coordinate defense. Guard cells (GCs), best known for regulating stomatal aperture, have poorly understood intrinsic immune roles. Here, we integrate single-cell and spatial analyses of *Arabidopsis thaliana* leaves infected with diverse phytopathogens, combining live and fixed imaging of transgenic immune reporters with single cell transcriptomic data. Powdery mildew infection triggers strong salicylic acid (SA) biosynthesis and transport in pavement cells, spreading to neighboring uninfected cells. In contrast, GCs fail to activate SA biosynthesis or SA-responsive genes. These cell-type-specific immune programs distinguishing GC and pavement cells are conserved across infections by the hemibiotrophic fungus *Colletotrichum higginsianum* and the bacterial pathogen *Pseudomonas syringae*. Despite impaired SA signaling and response, GCs display rapid calcium influx and pronounced reactive oxygen species bursts, transmitting immune signals to adjacent pavement cells via the apoplast. Notably, GCs are incompatible with adapted fungal pathogens and underwent hypersensitive cell death. Together, these findings uncover distinct immune programs among epidermal cell types, highlight GC-autonomous defense mechanisms, and provide a framework for understanding spatial immune coordination in plants.

Plants are constantly exposed to diverse microbial pathogens that can compromise plant growth and reproduction. Co-evolution of potentially pathogenic microbes and their hosts has shaped a highly resilient plant innate immune system[1]. In response to pathogen attack, plants initiate immune responses through a coordinated process involving immune recognition, signal integration, and defense execution[2]. Immune recognition is mediated by a diverse array of cell-surface and intracellular plant immune receptors[3]. Cell-surface pattern recognition receptors (PRRs) detect conserved microbial features known as pathogen-associated molecular patterns (PAMPs), triggering pattern-triggered immunity (PTI). PTI provides complete immunity against non-adapted pathogens and partial resistance to host-adapted pathogens. In turn, host-adapted pathogens deploy effector proteins into plant cells, where they interfere with immune responses[2]. These effectors are specifically recognized by intracellular nucleotide-binding leucine-rich repeat receptors (NLRs), which activate effector-triggered immunity (ETI)[2]. Recognition of potential pathogens by PRRs and NLRs initiates signal transduction cascades that culminate in immune activation[2,3]. PTI and ETI are not functionally isolated; instead, they reinforce one another and jointly mediate robust downstream immune responses, including calcium influx, reactive oxygen species (ROS) production, mitogen-activated protein kinase (MAPK) activation, phytohormone biosynthesis and signaling, and transcriptional reprogramming[3,4].

[1]Department of Biology, University of Saskatchewan, Collaborative Science Research Building, 112 Science Place, Saskatoon, SK S7N 5E2, Canada. [2]London Research and Development Centre, Agriculture & Agri-Food Canada, 1391 Sanford Street, London, ON N5V 4T3, Canada. ✉e-mail: yangdou.wei@usask.ca

Plant-pathogen interactions are highly dynamic processes shaped by the specific tissues and cell types of the host, as well as by the spatial distribution, developmental stage, and activity of the pathogen. Consequently, host-pathogen interactions at the cellular level are often heterogeneous and asynchronous, resulting in a mosaic of immune-activated and susceptible states. Conventional biochemical and molecular assays based on bulk tissue can obscure fundamental principles of cellular interactions. In contrast, recent advances in single-cell and spatial transcriptomics have begun to uncover distinct immune cell states, opening new avenues for cell type-specific investigations in plant immunity[3–6]. Despite these advances, fully elucidating the temporal dynamics of immune activation, signaling, cell-to-cell communication, and disease progression in the context of spatial heterogeneity among host cell types remains an urgent and unresolved challenge.

The leaf epidermis, the outermost layer of plant leaves, serves as a vital interface between the plant and its environment, and constitutes the first line of defense against pathogen invasion. The leaf epidermis comprises specialized cell types, including pavement cells and stomatal guard cells (GCs), which are essential for regulating gas exchange and influencing plant immunity[7–9]. A wide range of phytopathogens, such as bacteria, oomycetes and fungi, exploit stomatal openings as primary entry invasion routes[9]. In response, GCs can detect PAMPs or pathogen effectors through immune receptors, triggering stomatal closure to restrict pathogen entry and limit colonization[7–9]. However, some pathogens can counteract this response by secreting effector proteins or phytotoxins that disrupt GC function, leading to stomatal reopening and facilitating invasion[7–9]. To date, most research on stomatal immunity has focused on the regulation of stomatal aperture, revealing a complex signaling network involving calcium, abscisic acid (ABA), ROS, and defense hormones such as salicylic acid (SA)[7,9]. Despite these insights, the mechanisms underlying GC-autonomous immunity remain poorly understood.

Biotrophic powdery mildews are among the most widespread plant pathogens, typically colonizing the surface of plant tissue, primarily the epidermis. The infection process begins when conidiospores land on the plant surface and germinate, forming an appressorium that penetrates directly through the epidermal cell wall. This leads to the establishment of a unicellular haustorium, a specialized structure that facilitates nutrient uptake from the host within the infected epidermal cells. The infection process is highly coordinated, involving intricate, cell-specific interactions between host and pathogens[10]. In this study, we employed *Arabidopsis thaliana* (Arabidopsis) in both adapted and non-adapted powdery mildew pathosystems, integrating live and fixed tissue imaging with transgenic immune reporters to investigate immune responses, signaling dynamics, and cell-type-specific resistance or susceptibility in leaf epidermal pavement cells and GCs during infection. Our analysis provides single-cell and spatially resolved insights into SA biosynthesis and signaling, callose deposition, Ca²⁺ influx, and ROS production in response to powdery mildew infection. Intriguingly, while GCs do not engage in pathogen-induced SA synthesis and callose deposition, they exhibit pronounced Ca²⁺ influx and ROS accumulation during infection, revealing an inherent incompatibility with adapted fungal pathogens. Moreover, these cell-type-specific immune programs that distinguish GCs from pavement cells were conserved across infections by the hemibiotrophic fungus *Colletotrichum higginsianum* and the bacterial pathogen *Pseudomonas syringae*, as further supported by single-cell RNAseq analyses.

## Results

### Temporal and spatial induction of *ICS1* transcription during powdery mildew infection

Salicylic acid (SA) plays critical roles in plant defense against biotrophic pathogens. In Arabidopsis, the rate of pathogen-induced SA biosynthesis predominantly depends on transcription activity of *ISO-CHORISMATE SYNTHASE 1* (*ICS1*)[11]. To monitor this process, we used transgenic Arabidopsis plants harboring the reporter construct *pICS1::NLS-3×mVENUS*, which enables visualization of ICS1 promoter activity[12]. These plants were inoculated with conidiospores of the non-adapted barley-powdery mildew fungus *Blumeria hordei* (*B. hordei*)[13], and dynamics of *pICS1::NLS-3×mVENUS* expression were tracked via live-cell imaging during the course of infection. In general, *B. hordei* conidiospores germinate on the leaf surface of non-host Arabidopsis and form an appressorium, which develops a thin penetration peg to pierce the leaf epidermal cell wall. Most penetration attempts are unsuccessful due to the formation of dense papillae beneath the attempted entry site. However, in fewer than 5% of cases, penetration succeeds, followed by haustorium formation within 24-h post-inoculation (hpi)[14]. The invaded epidermal cells rapidly undergo hypersensitive cell death, effectively halting the progression of powdery mildew infection in the non-host Arabidopsis[14,15].

Leaves inoculated with *B. hordei* were sampled at three time points: 18, 21, and 24 hpi. To enable precise localization of fluorescent signals to specific cells and cell types, the inoculated leaf tissues were stained with the red fluorescent cell wall stain propidium iodide (PI). For quantitative analysis of signal induction, fluorescence intensities were measured in both the nuclei and the cytoplasm of epidermal cells either attempted or penetrated by *B. hordei* (designated as "Patient 0", P), as well as in their successive neighboring epidermal cell layers. Cells in direct contact with "Patient 0" were defined as the first layer (L1), while cells adjacent to L1, but not directly bordering "Patient 0", constitute the second layer (L2), and so forth (Supplementary Fig. 1A). At 18 hpi, no detectable induction of NLS-3×mVENUS fluorescence was observed in either the nuclei or the cytoplasm of cells at the penetration site (Fig. 1A, B). By 21 hpi, however, visible induction of nuclear-associated fluorescent signals became apparent in "Patient 0" and adjacent L1 cells (Fig. 1A, B). This induction intensified by 24 hpi, reflecting a time-dependent increase in signal intensity at the fungal penetration sites (Fig. 1A, B). Notably, enhanced fluorescence was also detected in the cytoplasm of "Patient 0" and L1 cells. These observations indicate that fungal penetration attempts, even before successful host cell entry, can activate *ICS1* transcription in the surrounding leaf tissues.

At 24 hpi, a strong NLS-3×mVENUS fluorescence signal was observed in "Patient 0" cells exhibiting visible haustorial formation (Fig. 1C, D). Notably, fluorescence induction was also detected in adjacent L1 and L2 cells, forming a prominent patchy induction pattern in which nuclear signal intensity peaked at the center and gradually declined toward the periphery (Fig. 1C, D). In the central region of the patch, fluorescence was evident not only in the nuclei but also in the cytoplasm of L1 and L2 cells (Fig. 1C, D). Z-stack imaging further revealed that the elevated fluorescent signals extended both horizontally across multiple layers of epidermal cells surrounding "Patient 0", and vertically into a few mesophyll cells directly beneath it (Fig.1E–G, Supplementary Movie 1). To assess whether this pattern also occurs during compatible interactions, *ICS1* reporter plants were inoculated with the adapted powdery mildew *Erysiphe cichoracearum* (*E. cichoracearum*), which establishes a compatible interaction with Arabidopsis[16,17]. At 26 hpi, induction of both nuclear and cytoplasmic NLS-3×mVENUS signals was evident in *E. cichoracearum*-penetrated cells and their neighboring cells, resembling the *ICS1* induction pattern observed during *B. hordei* infection (Supplementary Fig. 1B–D). These findings indicate that *ICS1* expression is triggered in response to infection by both adapted and non-adapted powdery mildew fungi.

To validate the activity of the *pICS1::NLS-3×mVENUS* reporter line, we generated transgenic lines harboring *pICS1::ICS1-mSCARLET*, in which the native *ICS1* promoter drives the expression of ICS1 fused to the red fluorescent protein mSCARLET. As expected, the ICS1-mSCARLET signal co-localized with chloroplasts (Supplementary

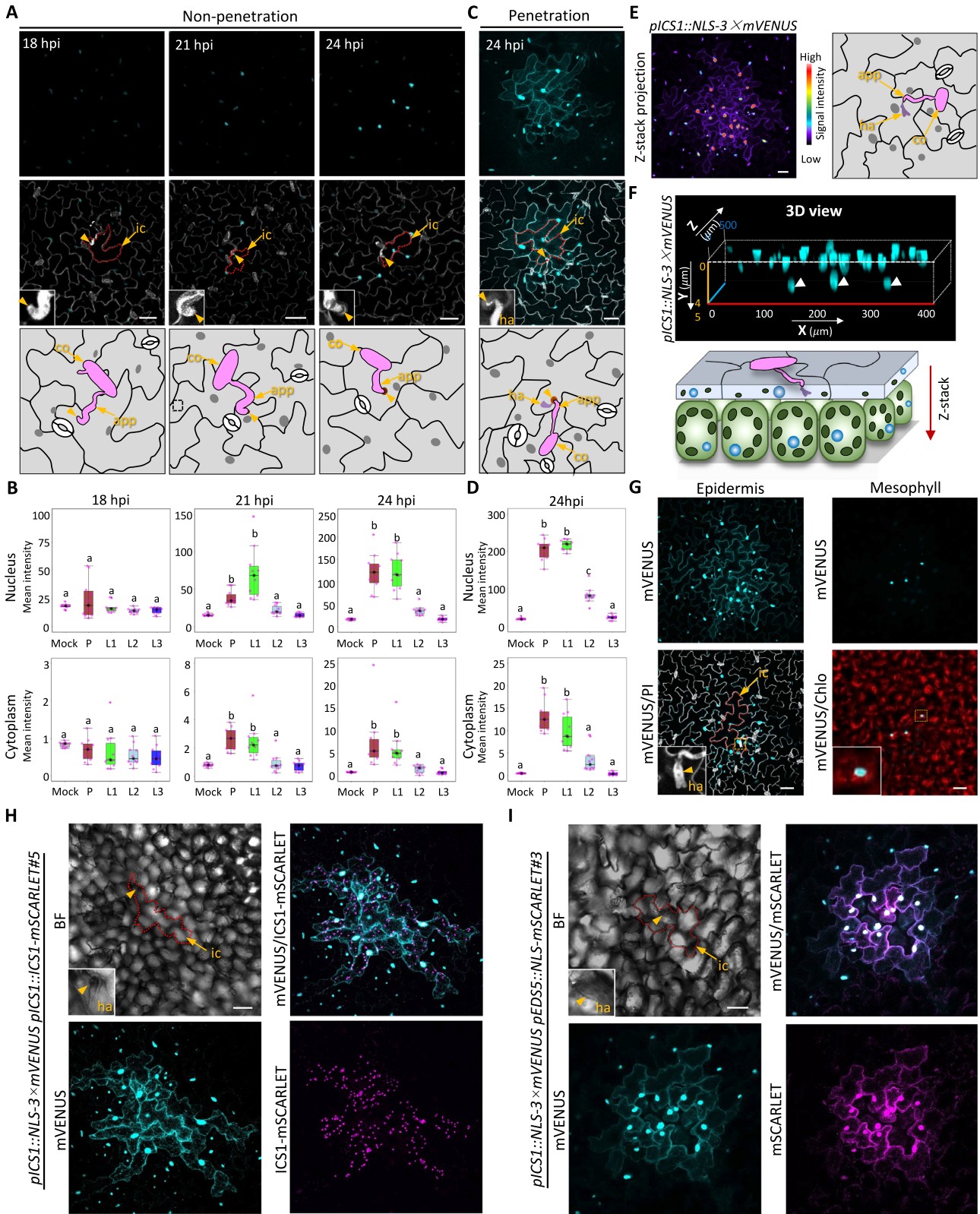

Fig. 2A), consistent with the plastidic localization of ICS1 protein[18]. We then performed a co-expression analysis by crossing this line with the *pICS1::NLS-3 × mVENUS* reporter line. Simultaneous visualization of nuclear-localized NLS-3 × mVENUS and plastid-localized ICS1-mSCAR-LET revealed complete spatial overlap of expression signals in *B. hordei*-infected cells and their immediate neighboring cells (Fig. 1H, Supplementary Fig. 2B).

In Arabidopsis, both *ICS1* and *ENHANCED DISEASE SUSCEPTIBILITY 5 (EDS5)* are crucial for SA accumulation in response to pathogen infection. ICS1 catalyzes the conversion of chorismate to iso-chorismate, while EDS5 facilitates the subsequent transport of iso-chorismate from the chloroplast to the cytosol. The expression of these genes is tightly regulated and follows a similar pattern[19]. To investigate whether *ICS1* and *EDS5* are co-expressed in a

**Fig. 1 | Spatiotemporal expression patterns of salicylic acid biosynthesis genes in *Arabidopsis* leaves in response to infection by the non-adapted powdery mildew *Blumeria hordei*.** **A** Confocal micrographs showing representative expression patterns of *pICS1::NLS-3 × mVENUS* in *Arabidopsis* leaf tissues surrounding *B. hordei*-attempted penetration sites at 20, 21, and 24 h post inoculation (hpi). Schematic illustrations (bottom panels) provide close-up views of the penetration sites. Fluorescent mVENUS signals are displayed as maximal intensity projections of Z-stacks. At least ten single-infection sites were analyzed at each time point. Cell walls and fungal structures stained with propidium iodide (PI) are shown in white. Infected cells are outlined with red dashed lines. Infection sites, indicated by arrowheads, are shown in enlarged views. ic, infected cell; co, conidiospore; app, appressorium; ha, haustorium. Scale bars, 50 μm. **B** Quantitative analysis of nuclear and cytoplasmic mVENUS signal intensities at *B. hordei*-attempted penetration sites at 20, 21, and 24 hpi in (**A**). The average mean intensity of Lx nuclei or cytoplasm was calculated as: sum (mean intensity of individual L1 nucleus or cytoplasm)/ number of Lx cells, where Lx = L1, L2, and L3. **C** Confocal micrographs showing representative expression patterns of *pICS1::NLS-3 × mVENUS* in *Arabidopsis* leaf tissues surrounding *B. hordei*-penetrated sites at 24 hpi. Schematic illustrations (bottom panels) provide close-up views of the penetration sites. Fluorescent mVENUS signals are displayed as maximal intensity projections of Z-stacks. At least ten single-infection sites were analyzed. Cell walls and fungal structures stained with propidium iodide (PI) are shown in white. Infected cells are outlined with red dashed lines. Infection sites, indicated by arrowheads, are shown in enlarged views. ic, infected cell; co, conidiospore; app, appressorium; ha, haustorium. Scale bars, 50 μm. **D** Quantitative analysis of nuclear and cytoplasmic mVENUS signal intensities at *B. hordei*-penetrated sites at 24 hpi in (**C**). The average mean intensity of Lx nuclei or cytoplasm was calculated as: sum (mean intensity of individual L1 nucleus or cytoplasm)/number of Lx cells, where Lx = L1, L2, and L3. **E**–**G**, Spatial distribution of *pICS1::NLS-3 × mVENUS* expression in leaf tissues surrounding a *B. hordei*-penetrated site at 24 hpi. **E** mVENUS signal intensity across the epidermal and

mesophyll layers, visualized using a Z-stack maximum intensity projection and a rainbow2 scale (Zeiss). Scale bar, 50 μm. A schematic (right panel) illustrates a magnified view of the penetration site. **F** Side view of mVENUS signals in epidermal and mesophyll layers. Arrowheads indicate mesophyll cell nuclei. The accompanying schematic (bottom panel) represents the same perspective. Epidermal cells are depicted in gray; mesophyll cells in light green; chloroplasts as dark green ovals; and nuclei in blue. *B. hordei* conidiospores and appressoria are shown in pink, and the haustoria in dark purple. **G** Top-down view of mVENUS expression in the epidermal and mesophyll layers. mVENUS signals are shown in cyan; cell walls and fungal structures stained with PI appear in white; and chlorophyll autofluorescence is shown in red. Infected cells are outlined with red dashed lines. Scale bar, 50 μm. **H** Co-expression patterns of *pICS1::NLS-3 × mVENUS* and *pICS1::ICS1-mSCARLET* in leaf tissues surrounding the *B. hordei*-penetrated site at 24 hpi. Fluorescent images are presented as maximal intensity projections of Z-stacks. mVENUS signals appear in cyan, and mSCARLET signals in magenta. Infected cells are outlined with red dashed lines, and infection sites indicated by arrowheads are shown in magnified insets for clarity. Scale bars, 50 μm. **I** Co-expression patterns of the double reporter constructs *pICS1::NLS-3 × mVENUS* and *pEDS5::NLS-mSCARLET* in leaf tissues surrounding the *B. hordei*-penetrated site at 24 hpi. Images are shown as maximal intensity projections of Z-stacks. mVENUS signals are depicted in cyan and mSCARLET in magenta. Infected cells are outlined with red dashed lines, and infection sites indicated by arrowheads are magnified enlarged for view. Scale bars, 50 μm. **B**, **D** All boxplots show the median (center line, *n* = 10), second to third (25% to 75%) quartiles (box), minimum and maximum values (whiskers). The small letters a to c denote statistically significant differences between means with one-way ANOVA, followed by the post hoc Tukey multiple comparison tests ($p < 0.05$). Exact *p*-values are shown in the Source data. Source data are provided as a Source Data file. **E**–**I** Experiments were repeated three times, with at least ten single-infection sites examined.

spatiotemporally coordinated manner, we generated two EDS5 reporter lines (*pEDS5::NLS-mSCARLET#3* and *pEDS5::NLS-mSCARLET#9*) and crossed them with the *pICS1::NLS-3 × mVENUS* reporter to produce dual-reporter lines. As anticipated, both reporters exhibited identical induction patterns at the site of the *B. hordei* infection at 24 hpi (Fig. 1I, Supplementary Fig. 3A). Together, these results demonstrate that the established reporter lines enable precise, spatiotemporal visualization of SA-biosynthesis and accumulation at the single-cell resolution in leaf tissues in response to the pathogen attack. They further reveal that the pathogen-induced SA biosynthesis is rapidly initiated in infected cells prior to pathogen entry (Supplementary Fig. 3B).

## Pathogen-induced SA biosynthesis relies on Ca²⁺ signaling

Pathogen-induced *ICS1* expression and SA biosynthesis are controlled by a complex network of positive and negative regulators[3,20]. Among these, Ca²⁺ signaling plays a pivotal role. Upon pathogen attack, an influx of Ca²⁺ from the apoplast into the cytosol is triggered, and this Ca²⁺ signal is crucial for activating *ICS1* expression[21,22]. To investigate the requirement of Ca²⁺ signaling, we treated *pICS1::NLS-3 × mVENUS* reporter plants with LaCl₃, a broad-spectrum Ca²⁺ channel blocker[23]. This treatment completely abolished the characteristic patchy induction of the reporter at *B. hordei*-infected sites (Fig. 2A). We next examined the effects of additional Ca²⁺ signaling antagonists commonly used in plants[24]. Interestingly, treatments with erythrosin B or eosin Y, antagonists of P-type Ca²⁺-ATPase that mediate Ca²⁺ efflux, similarly suppressed the patchy induction pattern of *pICS1::NLS-3 × mVENUS* at 24 hpi following *B. hordei* infection (Supplementary Fig. 4). These results indicate that perturbation of Ca²⁺ fluxes disrupts pathogen-triggered *ICS1* expression, underscoring the essential role of Ca²⁺ homeostasis and signaling in this process. To further assess the impact of Ca²⁺ availability, we grew the dual-reporter line (*pICS1::NLS-3 × mVENUS pEDS5::NLS-mSCARLET*) in hydroponic solutions containing various concentrations of CaCl₂ (0.025 mM, 0.1 mM, and 0.5 mM) for one week. Following *B. hordei* inoculation, plants grown in 0.5 mM Ca²⁺ displayed strong induction of both *ICS1* and *EDS5* at 24 hpi,

whereas those cultured in 0.1 mM Ca²⁺ showed a weaker response (Fig. 2B). Strikingly, no induction of reporter signal was detected in plants grown in 0.025 mM Ca²⁺ (Fig. 2B). These findings indicate that a threshold level of Ca²⁺, likely determined by the concentration gradient between the apoplast and the cytosol, is required to activate signaling pathways that drive the transcriptional induction of *ICS1* and *EDS5* in infected cells and their neighboring cells during pathogen attack.

CALMODULIN-BINDING PROTEIN 60-LIKEg (CBP60g) and SAR-DEFICIENT 1 (SARD1), two closely related transcription factors, function as master positive regulators of SA biosynthesis by directly activating the expression of *ICS1* and *EDS5*. Mutations in both *cbp60g* and *sard1* completely abolished *ICS1* induction and SA accumulation in response to bacterial pathogens[25]. Conversely, three calmodulin (CaM)-binding transcription activators (CAMTAs) serve as negative regulators by repressing the expression of *CBP60g* and *SARD1*, thereby limiting SA biosynthesis. In the *camta1/2/3* triple mutant, this repression is relieved, resulting in elevated *CBP60g* and *SARD1* expression and increased SA levels[26]. To visualize *ICS1* expression in these regulatory backgrounds, we introduced the *pICS1::NLS-3 × mVENUS* reporter into both *cbp60g sard1-2* double mutant and *camta1/2/3* triple mutant backgrounds via genetic crosses. Under non-infected conditions, the reporter exhibited low basal activity in both wild-type (WT) and *cbp60g sard1-2* plants. In contrast, strong fluorescent signals were detected in *camta1/2/3* plants, indicating constitutive activation of *ICS1* expression (Fig. 2C, Supplementary Fig. 5A–C). Upon *B. hordei* infection at 24 hpi, NLS-3 × mVENUS signals were strongly induced at the infection sites in WT plants but remained at basal levels in the *cbp60g sard1-2* mutants. Notably, *camta1/2/3* plants exhibited no further increase in fluorescence upon infection, suggesting that *ICS1* expression was already saturated in this mutant in the absence of pathogen challenge (Fig. 2D, Supplementary Fig. 5D). Our live-cell imaging results support previous findings that SA biosynthesis in response to pathogen attack is tightly controlled by Ca²⁺/CaM-regulated transcription factors CBP60g and CAMTAs (Supplementary Fig. 5E)[21,27].

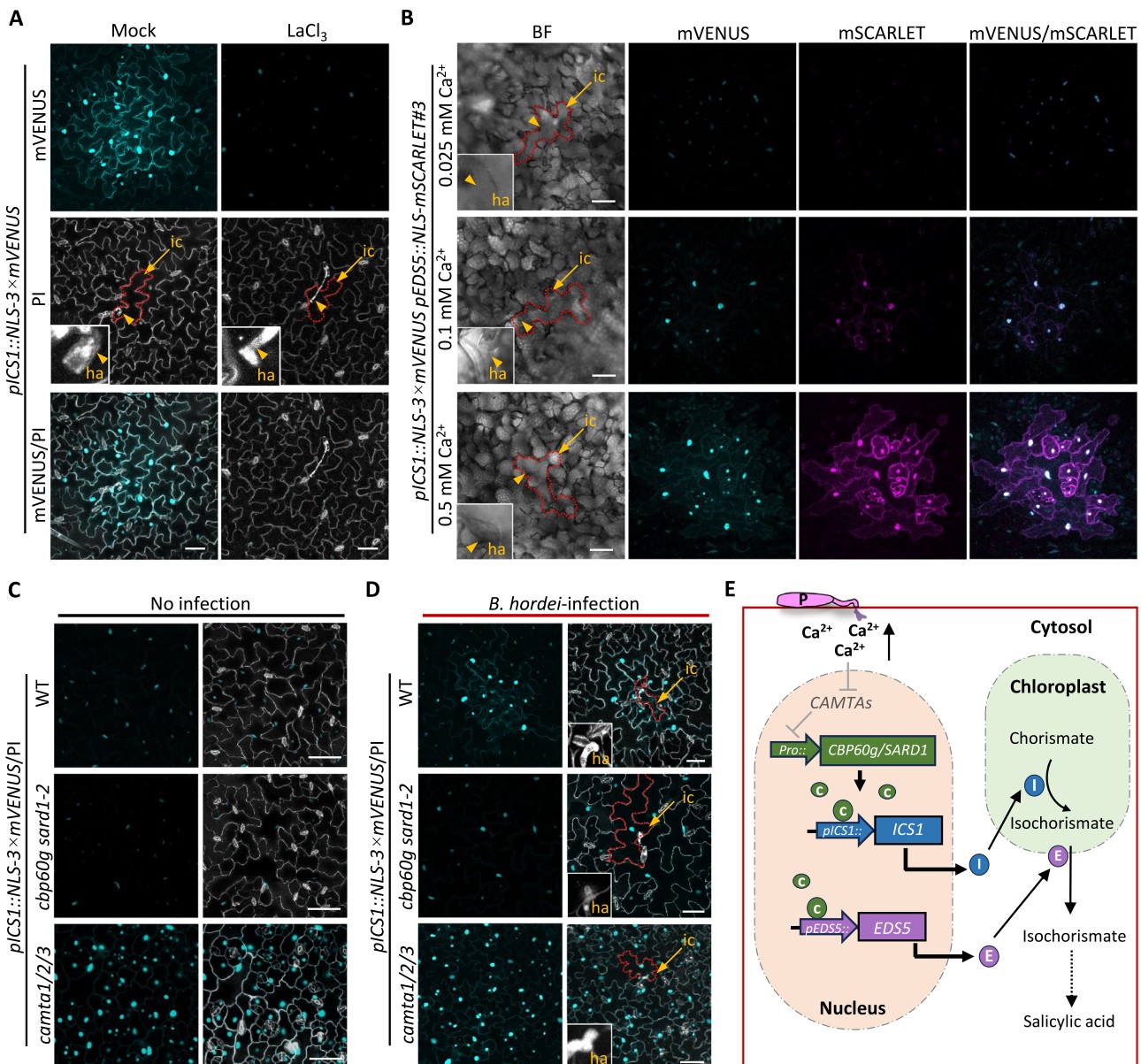

**Fig. 2 | Pathogen-induced salicylic acid biosynthesis is tightly regulated by Ca²⁺-dependent signaling pathways. A** Representative confocal image showing the expression pattern of the *ICS1* reporter at 24 hpi with *B. hordei*, with or without LaCl₃ treatment. Images are displayed as maximal intensity projections of Z-stacks. Plants infiltrated with water served as the mock control. Experiments were repeated three times, with at least ten single-infection sites examined per treatment in each replicate. mVENUS signals are shown in cyan; cell walls and fungal structures stained with PI appear in white. Infected cells are outlined with red dashed lines. Infection sites, indicated by arrowheads, are shown in enlarged views. ic, infected cell; ha, haustorium. Scale bars, 50 μm. **B** Representative images showing the expression patterns of *ICS1* and *EDS5* in the upper leaf epidermis of the double reporter plants (*pICS1::NLS-3 × mVENUS pEDS5::NLS-mSCARLET*) grown in hypotonic culture supplemented with varying concentrations of CaCl₂ (0.025, 0.1, and 0.5 mM) at 24 hpi with *B. hordei*. Fluorescent images are presented as maximal intensity projections of Z-stacks. Experiments were repeated three times. For each biological replicate, at least five single-infection sites were examined per CaCl₂ concentration. mVENUS signals are shown in cyan; mSCARLET signals in magenta. Scale bars, 50 μm. **C** Representative images showing the expression pattern of *ICS1* in non-inoculated WT, *cbp60g sard1-2*, and *camta1/2/3* mutant plants. At least ten plants of each genotype were examined. Images are displayed as maximal intensity projections of Z-stacks. mVENUS signals are shown in cyan; cell walls and fungal structures stained with PI in white. Scale bars, 50 μm. **D** Representative images showing pathogen-induced expression patterns of *ICS1* in WT, *cbp60g sard1-2*, and *camta1/2/3* mutant plants at 24 hpi with *B. hordei*. At least ten single-infection sites were analyzed for each genotype. Images are displayed as maximal intensity projections of Z-stacks. mVENUS signals are shown in cyan; cell walls and fungal structures stained with PI appear in white. Scale bars, 50 μm. **E** Model summarizing the regulation of *ICS1* and *EDS5* in pathogen-induced salicylic acid biosynthesis and accumulation. P, Pathogen; C, CBP60g/SARD1; I, ICS1; E, EDS5.

Taken together, these results suggest that *B. hordei* infection triggers a localized Ca²⁺ influx, which modulates CAMTA activity and subsequently upregulates the transcription of *CBP60g* and *SARD1* during pathogen-induced SA biosynthesis (Fig. 2E).

Multiple mechanisms contribute to the feedback amplification of SA biosynthesis and signaling, involving both positive and negative regulatory pathways[28]. To further explore this regulatory network, we introduced the *pICS1::NLS-3 × mVENUS* reporter into the *ics1* mutant (*sid2-1*) and generated dual *pICS1::NLS-3 × mVENUS pEDS5::NLS-mSCARLET* reporters in the SA-deficient *NahG* background through genetic crossing. Notably, following *B. hordei* infection, the reporters displayed comparable induction patterns in WT, *sid2-1*, and *NahG*

backgrounds (Supplementary Figs. 6, 7). These results suggest that *B. hordei*-induced activation of *ICS1* and *EDS5* occurs independently of SA or its precursor chorismate accumulation.

## Guard cells do not engage in SA biosynthesis and signaling in response to *B. hordei* infection

The Arabidopsis leaf epidermis comprises pavement cells and specialized cell types, including GCs and trichomes. Notably, in *pICS1::NLS-3 × mVENUS pEDS5::NLS-mSCARLET* dual reporter plants, fluorescent signals were undetectable in GCs adjacent to *B. hordei*-penetrated pavement cells, even though surrounding pavement cells showed strong induction of both mVENUS and mSCARLET signals (Fig. 3A). To visualize GC nuclei alongside reporter expression, we crossed the *pICS1::NLS-3 × mVENUS* reporter line with a nuclear marker line expressing *CENH3-mRFP* (Fig. 3B), and the *pEDS5::NLS-mSCARLET* line with an N7-GFP nuclear marker (Fig. 3C). Neither mVENUS nor mSCARLET fluorescence was detected in the nuclei of GCs adjacent to *B. hordei*-penetrated pavement cells (Fig. 3B, C), confirming that both *ICS1* and *EDS5* are not transcriptionally activated in these GCs. To determine whether this lack of expression results from a failure in signal transmission from the infected pavement cells or from an intrinsic inability of GCs to activate these genes, we examined *pICS1::NLS-3 × mVENUS* expression in GCs directly infected by *B. hordei*. No mVENUS signal was detected in the infected GCs (Fig.3D). Interestingly, the surrounding pavement cells exhibited strong induction of *pICS1::NLS-3 × mVENUS* (Fig.3D), suggesting that while GCs do not activate SA biosynthesis via ICS1 in response to *B. hordei* infection, they may nonetheless contribute to local immunity by releasing signals that activate defense responses in neighboring pavement cells.

In Arabidopsis, SA functions as an endogenous signal that activates the expression of downstream defense genes, including those encoding pathogenesis-related (PR) proteins. Among these, *PR1* and *PR4* are well-characterized SA-inducible genes[29,30]. To investigate whether GCs respond to SA, we treated the *PR1* reporter plants (*pPR1::NLS-3 × mVENUS CENH3-mRFP*) and the *PR4* reporter plants (*pHEL::NLS-3 × mVENUS CENH3-mRFP*) with exogenous SA[12]. Reporter analysis revealed that promoter activities of *PR1* and *PR4* were induced in pavement cells but not in GCs, as no fluorescent signal was detected in GCs (Supplementary Fig. 8A, B). Similar expression patterns were observed in *PR1* and *PR4* reporter plants inoculated with *B. hordei* (Supplementary Fig. 8A, B). These findings indicate that GCs are unresponsive to SA, supporting the conclusion that they are inactive in both SA biosynthesis and SA-mediated signaling and response during powdery mildew infection.

To determine whether this differential responsiveness between pavement cells and GCs is specific to SA-mediated immunity or reflects a broader divergence in immune activation, we analyzed the expression patterns of additional immune marker genes using the reporter lines *pFRK1::NLS-3 × mCITRINE CENH3-mRFP* and *pLipoP1::NLS-3 × mCITRINE CENH3-mRFP*, which track the transcriptional activity of *FRK1* and *LipoP1*, respectively. *FRK1* encodes a receptor-like protein of unknown function that is strongly and rapidly induced by PAMPs[31], while *LipoP1* encodes a putative membrane lipoprotein also implicated in early immune responses[6]. At 24 hpi with *B. hordei*, both *FRK1* and *LipoP1* exhibited expression patterns similar to that of the *pICS1::NLS-3 × mVENUS* and *pEDS5::NLS-mSCARLET* reporters, with induced signals observed in pavement cells surrounding the *B. hordei*-penetrated cells (Fig. 3E, F). Notably, expression of *pFRK1::NLS-3 × mCITRINE* was undetectable in GCs adjacent to infection sites (Fig. 3E). In contrast, *pLipoP1::NLS-3 × mCITRINE* showed strong nuclear-localized signals in GCs neighboring the penetrated cells (Fig. 3F), suggesting that immune signals originating in infected pavement cells may be translocated into adjacent GCs, selectively activating specific immune pathways.

Together, these findings reveal that GCs exhibit immune responses that are distinct from those of pavement cells. While GCs are deficient in SA biosynthesis and related immune defense, they retain the capacity to activate alternative immune pathways, indicating a unique and selective immune transcriptional profile within the leaf epidermis.

## Comparative transcriptional dynamics of pavement and guard cell immune responses to *Colletotrichum higginsianum* and *Pseudomonas syringae*

To investigate whether the distinct cell type-specific immune responses of pavement and guard cells represent a conserved mechanism against both fungal and bacterial pathogens, we analyzed the *pICS1::NLS-3 × mVENUS* reporter responses to the hemibiotrophic fungal pathogen *Colletotrichum higginsianum* (*C. higginsianum*)[32] and the bacterial pathogen *Pseudomonas syringae* pv. *tomato* (*P. syringae*).

Following *C. higginsianum* infection, pronounced reporter fluorescence was detected in pavement cells at 48 hpi, specifically in penetration-associated cells and their adjacent neighbors, while GCs exhibited no detectable induction of *ICS1* reporter activity (Fig. 4A). Tang et al. employed single-cell transcriptomics to characterize heterogeneous immune responses of Arabidopsis leaf tissues to *C. higginsianum*, uncovering vasculature-enriched specific immune receptor expression and cellular processes triggered by direct pathogen contact[4]. We reanalyzed their dataset to identify *C. higginsianum*-induced genes expressed in pavement and guard cells. The Venn diagram revealed a distinct separation of induced gene sets, with 558 genes uniquely upregulated in pavement cells, 395 in GCs, and only 213 shared between the two cell types (Fig. 4B). Gene ontology (GO) analysis revealed functional specialization, with pavement cell-specific genes enriched in chorismate biosynthetic, positive regulation of gibberellin biosynthesis, intralumenal vesicle formation, and post-translational protein targeting to the ER, reflecting activation of SA biosynthesis and secretory pathways (Fig. 4C). In contrast, GC-specific genes were significantly enriched in aspartate family amino acid catabolic processes, L-leucine catabolism, sucrose starvation response, and the reductive pentose-phosphate cycle, indicative of metabolic reprogramming toward energy redistribution and enhanced stress adaption (Fig. 4C). Genes commonly induced in both cell types were mainly associated with indolalkylamine biosynthesis, lytic vacuole organization, nitrogen compound detoxification, and assembly of mitochondrial respiratory chain complex II (Fig. 4C). Collectively, these results indicate that *C. higginsianum* infection elicits distinct immune programs in pavement and guard cells, with pavement cells activating SA-related pathways and guard cells undergoing metabolic reprogramming for stress adaptation.

To determine whether this cell type-specific immune pattern also occurs during bacterial infection, we infiltrated leaves of *pICS1::NLS-3 × mVENUS* reporter plants with the avirulent *P. syringae* strain AvrRpt2. At 12 hpi, strong *ICS1* reporter activity was observed in the *P. syringae*-infiltrated leaf area (Fig. 4D). We further challenged *pICS1::NLS-3 × mVENUS pICS1::ICS1-mSCARLET* plants with a panel of *P. syringae* strains, including the virulent strain *Pst* DC3000 and the avirulent strains *AvrRpt2*, *hrcC*, *hrpA*, and *hrpS*[17,33]. In all cases, fluorescence from both *NLS-3 × mVENUS* and *ICS1-mSCARLET* reporters was predominantly confined to pavement cells, whereas GCs exhibited no detectable reporter activity or *ICS1* expression (Fig. 4E, Supplementary Fig. 9). These findings indicate that *ICS1* expression remains strictly confined to pavement cells during bacterial infection. Recent single-cell RNA sequencing studies have elucidated the molecular interactions between Arabidopsis and *P. syringae*[5,6,34]. Among these, Delannoy et al. identified distinct yet interconnected defense programs across major Arabidopsis leaf cell types, highlighting coordinated transcriptional reprogramming that distinguishes cell-autonomous from non-

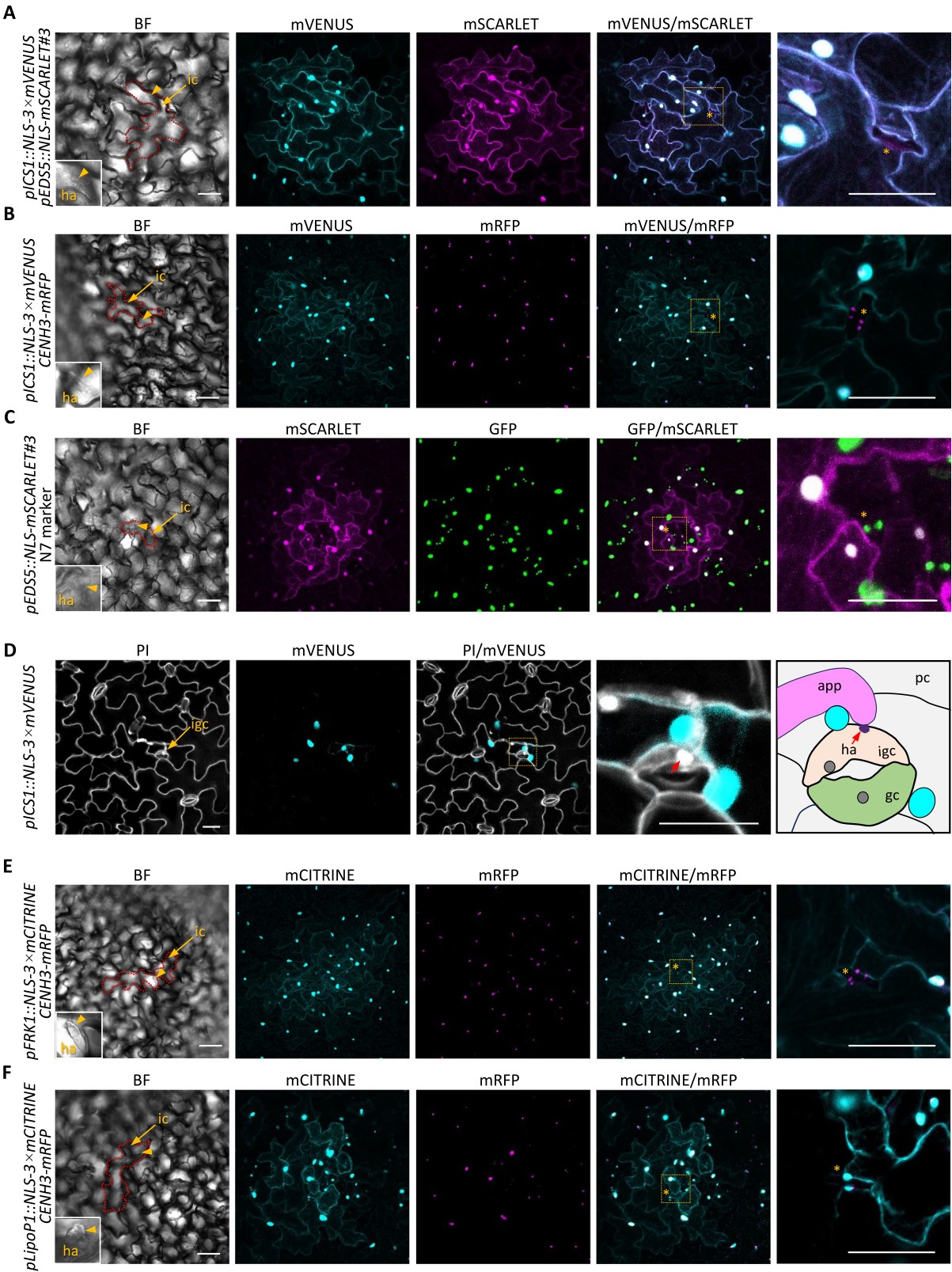

cell-autonomous immune responses to *P. syringae*[34]. Leveraging this dataset, we performed GO enrichment analyses of genes induced in pavement cells and GCs, revealing a clear functional division: GCs were predominantly enriched for responses to water deprivation, desiccation, and fluid transport, whereas pavement cells were enriched for SA signaling, hormone-mediated defense processes, and oxidative stress (Fig. 4F). Consistently, transcripts of *ICS1* and *PBS3* genes, the two

enzymes directly involved in SA biosynthesis, were absent in GCs (Fig. 4G, Supplementary Fig. 10A).

Together, these results demonstrate that pavement cells serve as the primary sites of pathogen-induced SA biosynthesis, while GCs prioritize metabolic adjustments and stress resilience, highlighting a conserved division of labor in epidermal immunity across fungal and bacterial infections.

**Fig. 3 | Distinct spatial expression patterns of immune genes in different epidermal cell types upon *B. hordei* infection. A–C** Representative images showing expression patterns of *ICS1* and *EDS5* in pavement and guard cells following *B. hordei* infection at 24 hpi. Images were acquired from transgenic lines co-expressing *pICS1::NLS-3 × mVENUS* and *pEDS5::NLS-mSCARLET* (**A**), *pICS1::NLS-3 × mVENUS* and *CENH3-mRFP* (**B**), and *pEDS5::NLS-mSCARLET* and *N7* (**C**). Fluorescent images are shown as maximal intensity projections of Z-stacks. At least ten single-infection sites were examined per reporter line. An enlarged view of guard cells adjacent to infected pavement cells is shown in the right panels. Infected cells are outlined with red dashed lines. mVENUS signals are depicted in cyan; mSCARLET and mRFP in magenta; GFP in green. Scale bars: 50 μm.
**D** Representative image showing the expression pattern of *pICS1::NLS-3 × mVENUS* in pavement cells surrounding a guard cell penetrated by *B. hordei*. At least ten single-infection sites were examined. The enlarged inset provides a close-up view of the infected guard cell. A diagram illustrates the details of the infection site. Observations were made at a minimum of ten single-infection sites. Cell walls and fungal structures stained with PI are shown in white. The red arrow indicates the haustorium. Scale bars, 20 μm. **E** Representative images showing the expression

pattern of the immune gene *FRK1* following *B. hordei* infection at 24 hpi. Images were obtained from transgenic plants co-expressing *pFRK1::NLS-3 × mCITRINE* and *CENH3-mRFP*. The enlarged inset shows a close-up of a guard cell adjacent to the penetrated pavement cell. Fluorescent signals are displayed as maximal intensity projections of Z-stacks. A minimum of ten single-infection sites were analyzed. The infected cell is outlined with a red dashed line. mCITRINE signals are shown in cyan; mRFP signals in magenta. Scale bars: 50 μm. **F** Representative images showing the expression pattern of the immune gene *LipoP1* following *B. hordei* infection at 24 hpi. Images were obtained from transgenic plants co-expressing *pLipoP1::NLS-3 × mCITRINE* and *CENH3-mRFP*. The enlarged inset highlights a guard cell adjacent to a penetrated pavement cell, showing a co-localization of *mCITRINE* and *mRFP* signals in guard cells. Fluorescent signals are presented as maximal intensity projections of Z-stacks. A minimum of ten single-infection sites were analyzed. The infected cell is outlined with a red dashed line. mCITRINE signals are shown in cyan; mRFP signals in magenta. Scale bars: 50 μm. **A–F** Guard cells adjacent to infected pavement cells are indicated by asterisks. ic infected cell; igc infected guard cell; pc pavement cell; app appressorium; ha haustorium.

## Guard cells exhibit a distinct cell-autonomous immune defense

Building on the distinct role of GCs in SA biosynthesis and immune gene expression, we further explored additional aspects of their cell-autonomous immune responses. Callose deposition, a well-known plant defense response against pathogen invasion, typically occurs at fungal penetration sites and is also associated with pathogen-induced cell death[35,36]. To assess callose accumulation in GCs during pathogen infection, we inoculated WT plants with *B. hordei* and stained infected leaf tissues with aniline blue (sirofluor) for callose visualization[37,38]. In pavement cells penetrated by *B. hordei*, callose was prominently deposited at the fungal entry site and around the affected cell (Fig. 5A, Supplementary Movie 2). High-resolution imaging further revealed that callose accumulation extended to adjacent pavement cells and underlying mesophyll tissues. Intriguingly, no callose deposition was observed in GCs adjacent to the *B. hordei*-penetrated pavement cell (Fig. 5A), suggesting a deficiency in pathogen-induced callose accumulation within GCs. To explore this further, we examined callose deposition in GCs directly targeted by *B. hordei*. Surprisingly, callose was undetectable within the penetrated GCs, neither at the site of fungal entry nor in surrounding regions. In contrast, neighboring pavement cells displayed clear callose accumulation at regions adjacent to the infected GCs (Fig. 5B, C, Supplementary Movie 3). These findings indicate that callose deposition is not a component of the GC-autonomous immune response, although GCs invaded by *B. hordei* may still contribute to non-autonomous immune signaling that triggers callose deposition in surrounding pavement cells.

Calcium and reactive oxygen species (ROS) are well-characterized second messengers in plant immunity, playing pivotal roles in stomatal signaling by regulating stomatal aperture[39]. To investigate $Ca^{2+}$ influx in GCs during pathogen infection, we expressed a cytosolic calcium sensor, MatryoshCaMP6[40], in WT plants and inoculated them with *B. hordei*. At 20 hpi, elevated cytosolic fluorescence was detected in GCs immediately adjacent to the penetrated pavement cell, whereas GCs located farther from the infected site did not show elevated $Ca^{2+}$ levels (Fig. 5D, E). These results indicate a localized and sustained increase in cytosolic $Ca^{2+}$ in GCs influenced by neighboring *B. hordei*-infected cells. Elevated cytosolic $Ca^{2+}$ activates the NADPH oxidases RBOHD and RBOHF either through direct $Ca^{2+}$ binding or via phosphorylation mediated by $Ca^{2+}$-dependent protein kinases (CPKs), thereby amplifying ROS production[41–43]. Consistent with this mechanism, genes associated with ROS generation were strongly induced in GCs upon *P. syringae* infection (Supplementary Fig. 10B). To assess ROS accumulation in GCs following infection, we inoculated the *pEDS5::NLS-mSCARLET* reporter line with *B. hordei* and infiltrated the infected leaves with $H_2$DCF-DA, a cell-permeable fluorescent probe that is deacetylated by intracellular esterases and oxidized by ROS, enabling

visualization of ROS accumulation in live GCs[44]. ROS accumulation was detectable in GCs surrounding *B. hordei*-attempted pavement cells as early as 18 hpi, preceding fungal penetration (Supplementary Fig. 11A, B). At 24 hpi, the induced NLS-mSCARLET signal in the *EDS5* reporter line marked the infected regions (Fig. 5F, G). While the NLS-mSCARLET signal itself was not observed in GCs, pronounced ROS accumulation was detected in GCs located within these *EDS5*-expressing regions. In contrast, GCs outside these regions exhibited no significant ROS accumulation (Fig. 5F, G). By 26 hpi, autofluorescence was observed in GCs adjacent to *B. hordei*-infected cells (Fig. 5H), suggesting the onset of hypersensitive cell death in these cells. Notably, infection by compatible *E. cichoracearum* induces higher $Ca^{2+}$ elevation and ROS accumulation in GCs adjacent to the infected pavement cell than in the surrounding pavement cells at 22–26 hpi (Supplementary Fig. 11C–F). These results suggest that GCs respond more vigorously to pathogen challenges than pavement cells, demonstrated by their more rapid and pronounced $Ca^{2+}$ elevation as well as heightened intracellular ROS accumulation.

We further investigated the interplay between SA signaling, ROS accumulation, and hypersensitive cell death. Arabidopsis plants expressing the bacterial salicylate hydroxylase gene *NahG* under the CaMV35S promoter accumulate little to no SA[45]. To assess the role of SA specifically in epidermal cells, we generated transgenic Arabidopsis lines expressing *NahG-mSCARLET* driven by the epidermis-specific *AtML1* promoter[46]. As expected, NahG-mSCARLET signals were restricted to the leaf epidermis (Supplementary Fig. 12A). These *pAtML1::NahG-mSCARLET* plants exhibited a similarly enhanced susceptibility to the adapted powdery mildew *E. cichoracearum* as plants expressing *CaMV35S::NahG* (Supplementary Fig. 12B). Following *B. hordei* challenge, epidermal cells of NahG-mSCARLET leaves penetrated by *B. hordei* displayed strong ROS accumulation, accompanied by cell autofluorescence, while neighboring cells showed callose deposition (Supplementary Fig. 12C–E). These results indicate that *B. hordei*-induced cell death in Arabidopsis is associated with ROS accumulation and autofluorescence and occurs independently of SA accumulation.

Together, these findings reveal that GCs exhibit a distinct, cell-autonomous immune response characterized by a deficiency in callose deposition, yet marked by sustained cytosolic $Ca^{2+}$ elevation and ROS accumulation, culminating in accelerated cell death (Fig. 5I).

## Guard cells are incompatible with host-adapted fungal pathogens

Previous studies have shown that GC death in both monocot and dicot species can arrest invasion of non-adapted fungal pathogens[47,48]. These findings led us to investigate how GCs respond to host-adapted

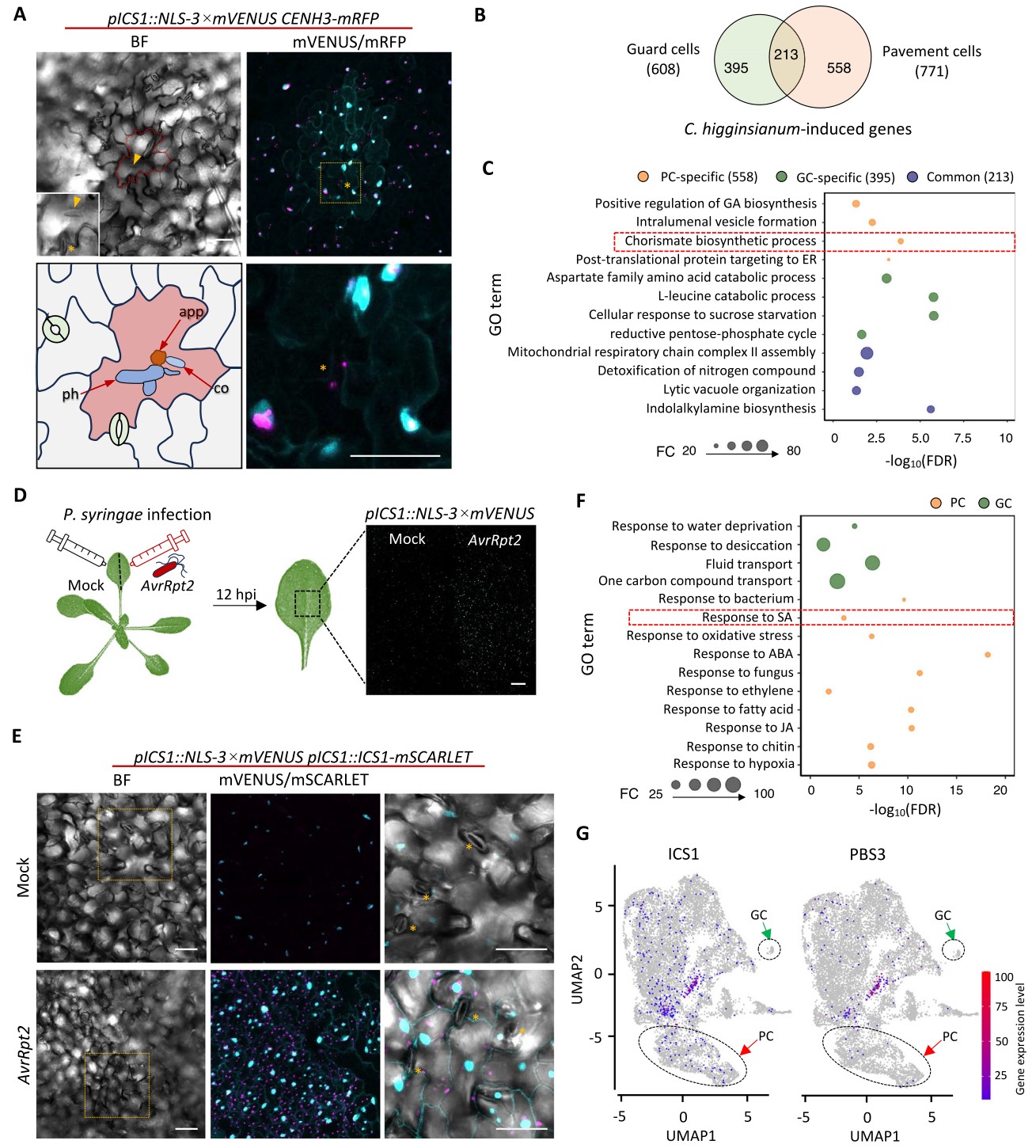

pathogens. We then inoculated the 14-day-old WT plants with the adapted powdery mildew *E. cichoracearum*, and fungal development was assessed at 72 hpi by confocal microscopy. In pavement cells, *E. cichoracearum* successfully penetrated via appressoria, establishing fungal colonies marked by the formation of mature haustoria with a globular central body and projecting lobes and the emergence of elongating secondary hyphae on the leaf surface (Fig. 6A, B, Supplementary Movie 4). In striking contrast, *E. cichoracearum* infection was arrested in GCs. Only haustorial primordia were observed, lacking the characteristic lobed morphology of mature haustoria, and no secondary hyphae developed from these infection sites (Fig. 6A, C, Supplementary Movie 5). These findings suggest that GCs are incompatible with this host-adapted powdery mildew pathogen.

Interestingly, callose deposition was observed specifically at the *E. cichoracearum* penetration sites within infected pavement cells, while it was barely present in the surrounding uninfected cells (Fig. 6A, B). In contrast, prominent callose accumulation was detected in pavement cells adjacent to infected GCs, whereas the GCs themselves showed little or no callose deposition at penetration sites (Fig. 6A, C).

To assess whether this GC incompatibility extends to other host-adapted fungal pathogens, we next examined infection by another compatible powdery mildew species, *Erysiphe cruciferarum* (*E. cruciferarum*)[49]. Despite forming more elaborate appressoria, *E. cruciferarum* exhibited a similar infection pattern closely resembling that of *E. cichoracearum*: GCs remained fully incompatible, displaying impaired haustorium formation and the absence of secondary hyphal

**Fig. 4 | Comparative transcriptional responses of pavement and guard cells to *Colletotrichum higginsianum* and *Pseudomonas syringae*. A** Representative images showing *pICS1::NLS-3 × mVENUS* CENH3-*mRFP* expression patterns in pavement and guard cells at 48 hpi with *C. higginsianum*. Fluorescent images are shown as maximal intensity projections of Z-stacks. The right panel presents an enlarged view of guard cells adjacent to infected pavement cells. At least ten single-infection sites were examined. The infected cell is outlined with a red dashed line. The guard cell adjacent to the infected pavement cell is indicated by an asterisk. mVENUS signals are depicted in cyan and mRFP in magenta. Asterisks mark guard cells immediately neighboring the infected pavement cell. Arrowheads indicate the primary hyphae within infected pavement cells. co, conidium; app, appressorium; ph, primary hyphae. Scale bars, 50 μm. **B** Venn diagram showing the number of induced genes in pavement and guard cells at *C. higginsianum* infection sites (data compiled from previous studies, Tang et al., 2023). The cutoff was set as log₂(FC) > 0.25, *p*-adj < 0.05. **C** Enrichment of biological processes for genes shown in (**B**). **D** Promoter activity of *ICS1* in three-week old *pICS1::NLS-3 × mVENUS* Arabidopsis leaves infiltrated with an avirulent *P. syringae* strain (*AvrRpt2*) or water (mock control) at 12 hpi. Experiments were repeated three times. Scale bar, 500 μM.

**E** Representative images showing *pICS1::NLS-3 × mVENUS* and *pICS1:ICS1-mSCARLET* expression patterns in pavement and guard cells following infiltration with *AvrRpt2*. Leaf tissues infiltrated with water served as mock controls. Guard cells are indicated by asterisks. The enlarged inset provides a close-up view of guard cells. Observations were made across at least three biological replicates. Fluorescent images are maximal intensity projections of Z-stacks. mVENUS signals are shown in cyan and mSCARLET in magenta. Scale bars, 50 μm. **F** Enrichment of biological processes for genes induced in pavement and guard cells upon *P. syringae* inoculation (data compiled from previous studies, Delannoy et al., 2023). **G** Cell-specific expression profiles of *ICS1* and *PBS3* encoding key enzymes in pathogen-induced SA biosynthesis in the UMAP projection. PC pavement cell; GC guard cell. **C, F** Functional enrichment analysis was performed using the PANTHER Classification System (overrepresentation test). Statistical significance was assessed using two-sided Fisher's exact tests by comparing the input gene set to the whole genome background. To account for multiple hypothesis testing, *p*-values were adjusted using the Benjamini-Hochberg false discovery rate (FDR) procedure. Categories with an FDR-adjusted *p*-value < 0.05 were considered statistically significant.

development (Supplementary Fig. 13A–C, Supplementary Movie 6). We further tested GC susceptibility to *C. higginsianum*. In pavement cells, infection proceeded through appressoria, forming primary hyphae by 3 dpi (Fig. 6D, Supplementary Movie 7), which subsequently developed into thin secondary hyphae by 4 dpi (Supplementary Fig. 14). In contrast, *C. higginsianum* infection in GCs was limited to a short initial hypha, which failed to progress (Fig. 6D, Supplementary Movie 7).

Collectively, these findings demonstrate that GCs are broadly incompatible with multiple host-adapted fungal pathogens, displaying distinct immune signatures compared to their compatible pavement cell counterparts.

## Discussion

By combining live and fixed tissue imaging with transgenic immune reporters, our study provides a high-resolution, single-cell, and spatial analysis of immune responses in Arabidopsis leaves during pathogen infection. We developed a suite of transgenic reporter lines targeting key components of the salicylic acid (SA) biosynthesis, transport, and signaling pathways to monitor immune dynamics during pathogen infection. Temporal and spatial tracking of pathogen-induced SA biosynthesis using the *pICS1::NLS-3 × mVENUS* reporter revealed a propagation of transcriptional activation that originated from infected cells and extended into several layers of neighboring uninfected pavement and mesophyll cells (Fig. 1A–F, Supplementary Fig. 1). The *pEDS5::NLS-mSCARLET* reporters exhibited a highly similar spatiotemporal expression pattern (Fig. 1I, Supplementary Fig. 3), indicating a coordinated activation of SA biosynthesis and transport at the cellular level. Recent advances in single-cell and spatial transcriptomics have revealed complex immune landscapes in Arabidopsis leaves during bacterial infection, including the identification of primary immune responder (PRIMER) cells that exhibit non-canonical immune signatures and are spatially confined to immune-active regions [5]. These PRIMER cells are surrounded by a bystander cell population that activates genes associated with long-distance cell-to-cell immune communication [5]. Our study expands this spatial framework to encompass diverse pathosystems, including both fungal and bacterial infections, suggesting that similar principles of spatial immune organization operate across different pathogen classes.

The observed spread of infection-induced transcriptional activation of SA-related genes and other immune-responsive genes from infected to neighboring uninfected cells raises key mechanistic questions: Which signaling molecules mediate this intercellular communication, and through what pathways is the signal transmitted? Our pharmacological assays using exogenous Ca²⁺ supplementation and the non-specific broad-spectrum Ca²⁺ channel blocker LaCl₃ (Fig. 2A,

B, Supplementary Fig. 4) demonstrate that Ca²⁺ signaling is required for the infection-induced expression of *ICS1* and *EDS5*. This response appears to depend on reaching a critical Ca²⁺ concentration threshold, either in the apoplast or within the cytosol. Intriguingly, among the various Ca²⁺ signaling antagonists commonly used in both animal and plant systems, only eosin Y and erythrosin B, two potent inhibitors of ACA-P₂B type Ca²⁺-ATPases [24], specifically suppress the infection-induced patchy expression patterns of *ICS1* and *EDS5* (Supplementary Fig. 4). Members of the autoinhibited Ca²⁺-ATPase (ACA) family belong to the P-type Ca²⁺-ATPases, a class of active transporters that utilize ATP to pump Ca²⁺ out of the cytosol into intracellular compartments or the apoplast, thereby restoring basal cytosolic Ca²⁺ levels after transient Ca²⁺ elevations. This resetting activity is essential for maintaining Ca²⁺ homeostasis and shaping the amplitude, duration, and spatial distribution of Ca²⁺ oscillations [50]. In Arabidopsis, ACA family members mediate active Ca²⁺ efflux into the apoplast, ER, and vacuole, thereby sustaining calcium gradients across subcellular compartments [50,51]. Several recent studies have shown that genetic disruption of these pathways leads to altered immune and stress responses [52–54]. Thus, ACA-type Ca²⁺-ATPases act as key "reset" components of the calcium signaling network, maintaining homeostasis while enabling precise control over the signaling dynamics after exposure to specific stimuli [50,51]. Our findings further suggest that ACA activity may contribute to infection-triggered SA biosynthesis and signaling through intercellular or cell-to-cell communication, although this mechanism warrants further genetic and molecular investigation.

Genetic evidence further supports the involvement of calcium signaling in regulating the infection-induced expression of *ICS1* and *EDS5*. Ca²⁺-dependent transcription factors CBP60g and CAMTAs act as positive and negative regulators, respectively, of this response (Fig. 2C, D, Supplementary Fig. 5) [21,26]. Interestingly, GCs adjacent to infected pavement cells exhibit pronounced cytosolic Ca²⁺ elevation (Fig. 5D, E); however, cytosolic Ca²⁺ levels alone did not account for the cell-type-specific activation of *ICS1* and *EDS5*. These data suggest that distinct Ca²⁺ influx dynamics or signaling profiles, rather than absolute Ca²⁺ levels, underlie the specialized immune responses of different cell types. Strikingly, pathogen-infected GCs could still transmit immune signals to neighboring pavement cells, inducing *ICS1* and *EDS5* expression and callose deposition. Given that mature GCs lack continuous plasmodesmata with adjacent pavement cells or between sister GCs [55,56], this immune signal must be relayed via the apoplast. Together, these findings support a model in which GCs act as specialized sensors that perceive pathogen invasion and coordinate defense responses in adjacent epidermal cells through apoplastic signaling.

Numerous studies investigating 'stomatal-based immunity' across diverse plant species have highlighted stomatal closure as the first line

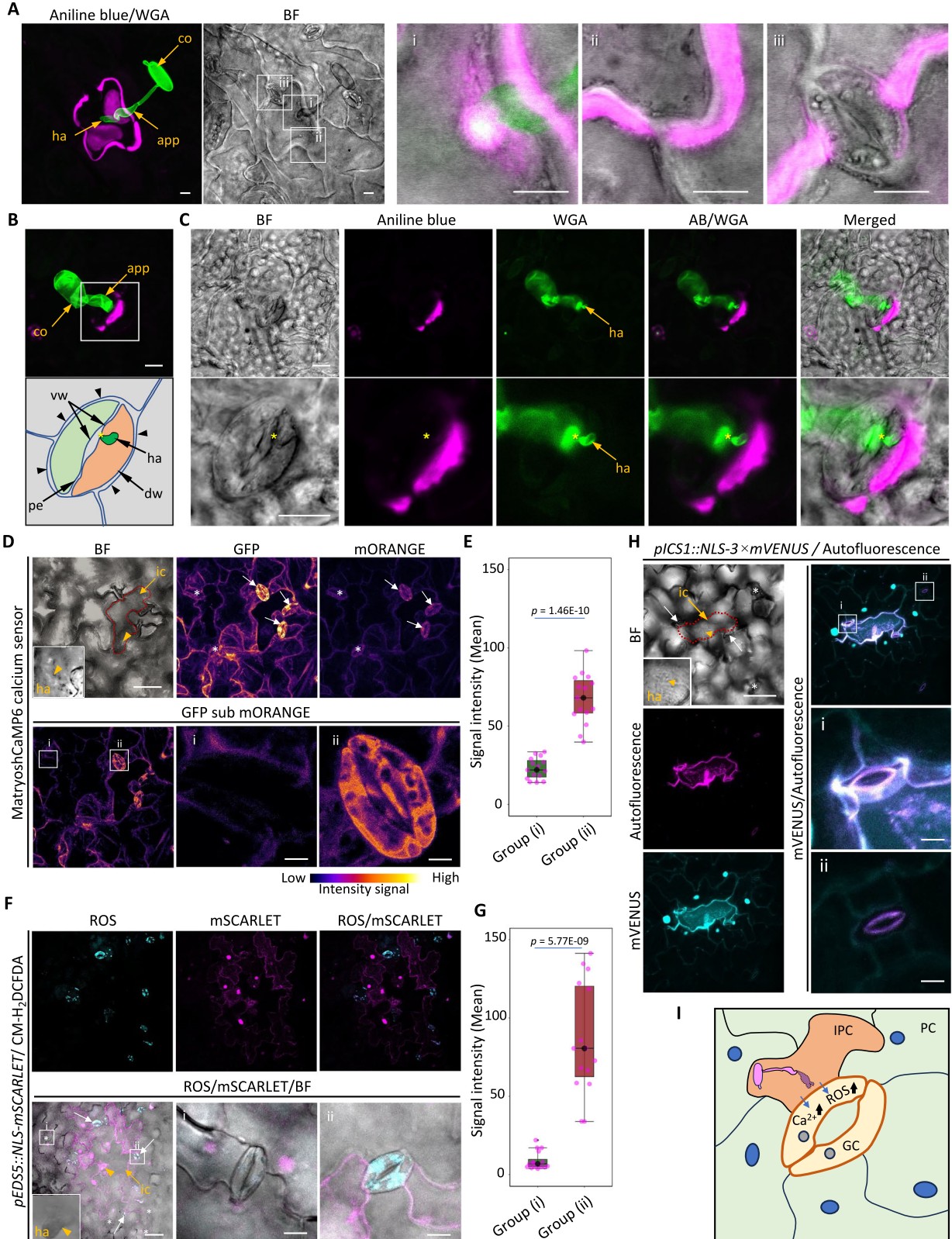

of defense against bacterial and filamentous pathogens[57–61]. Key signaling molecules, including abscisic acid (ABA), ROS, Ca²⁺, salicylic acid (SA) and jasmonic acid (JA), play a crucial role in regulating the stomatal aperture in response to pathogen challenge or abiotic stress. With respect to SA involvement in regulating stomatal immunity, the existing literature primarily examines SA function acting at the tissue or whole-organ level in modulating GC behavior, rather than exploring

the intrinsic capacity of GC themselves to synthesize or respond to SA[39,57]. Notably, GCs adjacent to infected pavement cells showed no activation of the *pICS1::NLS-3×mVENUS* or *pEDS5::NLS-mSCARLET* reporters (Fig. 3A–C), nor induction of SA-responsive *PR* genes (Supplementary Fig. 8). Even when fungal penetration occurred directly in GCs, reporter expression remained undetectable, whereas neighboring pavement cells displayed strong *ICS1* activation (Fig. 3D). Likewise,

**Fig. 5 | Cell-autonomous immune responses of guard cells during the *B. hordei* infection. A** Callose deposition associated with a *B. hordei*-penetrated pavement cell. *B. hordei*-inoculated leaf samples were fixed at 48 hpi. At least twenty single-infection sites were examined. Fungal structures were stained with wheat germ agglutinin conjugated to Alexa Fluor 488 (WGA, green), and callose deposition was visualized using aniline blue staining (magenta). A Z-stack projection obtained by confocal microscopy is shown. Insets display enlarged single–plane images of: (i) the *B. hordei*-penetration site, (ii) the boundary between the *B. hordei*-penetrated pavement cell and adjacent non-infected pavement cells, and (iii) the guard cell adjacent to the *B. hordei*-penetrated cell. Scale bars, 10 μm. **B, C** Callose deposition associated with a *B. hordei*-penetrated guard cell. At least ten single-infection sites were examined. **B** *B. hordei* penetration sites are marked with asterisks. The accompanying illustration details the stomatal structure of the infected guard cell. Anatomical features are labeled as follows: vw, ventral wall; dw, dorsal wall; pe, polar end of the ventral wall. Arrows indicate the walls of pavement cells surrounding the infected guard cell. **C** Insets provide magnified views of guard cell infection and associated callose deposition in the adjacent pavement cells. *B. hordei* penetration sites are marked with asterisks. Scale bars, 10 μm. **D** Cytosolic Ca$^{2+}$ dynamics detected by the calcium sensor MatryoshCaMP6 in *B. hordei* infected leaf tissue at 22 hpi. Fluorescent signals from GFP and mORANGE channels are displayed as maximum-intensity projections of Z-stacks. At least ten single-infection sites were examined. The infected cell is outlined with a red dashed line. White asterisks mark non-immediate neighboring guard cells, and white arrows point to immediate neighboring guard cells. Signal intensities for GFP and mORANGE are shown as the mpl-inferno LUT (ImageJ). Scale bars, 50 μm. Enlarged panels show a representative example of a non-immediate neighbor guard cell (i) and an immediate neighbor guard cell (ii). ic, infected cell; ha, haustorium. Scale bars (i-ii), 10 μm. **E** Boxplot showing the quantitative analysis of mean GFP/mORANGE signal intensities in two groups of guard cells as defined in (**D**). Group (i) represents non-immediate neighboring guard cells, and Group (ii) represents immediate

neighboring guard cells of the infected pavement cell. **F** Accumulation of reactive oxygen species (ROS) detected by CM-H$_2$DCFDA in *B. hordei* infected leaf tissues of *pEDS5::NLS-mSCARLET* plants at 24 hpi. CM-H$_2$DCFDA staining and mSCARLET fluorescence are shown as maximal intensity projections of Z-stacks. A minimum of ten single-infection sites were examined. ROS signals are displayed in cyan, and mSCARLET signals in magenta. White asterisks mark non-immediate neighboring guard cells (i) of the infected pavement cell, white triangles point to immediate neighboring guard cells (ii) of the infected cell. Scale bars, 50 μm. Enlarged views show representative examples of a non-immediate neighbor guard cell (i) and an immediate neighbor guard cell (ii). Scale bars (i-ii), 10 μm. **G** Boxplot showing quantitative analysis of mean ROS signal intensities in two groups of guard cells (Group I and Group II) as defined in (**F**). Group (i) represents non-immediate neighboring guard cells, and Group (ii) represents immediate neighboring guard cells of the infected pavement cell. **H** Autofluorescence in *B. hordei*-penetrated pavement cell and adjacent guard cells. Leaf tissues from *pICS1::NLS-3× mVENUS* plants were analyzed at 26 hpi. A minimum of ten single-infection sites were examined. Autofluorescence signals are shown in magenta, mVENUS signals in cyan. White asterisks mark guard cells (i) located distal to the infected pavement cell, while white triangles denote immediate neighbor guard cells (ii). Scale bars, 50 μm. Enlarged views show representative examples of a non-immediate neighbor guard cell (i) and an immediate neighbor guard cell (ii). Scale bars (i-ii), 10 μm. **I** Schematic diagram illustrating elevated cytosolic Ca$^{2+}$ and ROS levels, along with autofluorescence, in guard cells in response to *B. hordei* infection. PC, pavement cell; IPC, infected pavement cell; GC, guard cell. Blue circles represent nuclei of pavement cells exhibiting *B. hordei*-induced activation of the *pICS1::NLS-3× mVENUS* reporter. Gray circles indicate GC nuclei lacking *ICS1* promoter activity. **E, G** All boxplots show the median (center line, *n* = 15), second to third (25% to 75%) quartiles (box), minimum and maximum values (whiskers) of measurement. Statistical significance was assessed using a two-tailed unpaired Student's *t*-test; All *p*-values are indicated. Source data are provided as a Source Data file.

---

the early immune gene *FRK1*, typically upregulated by PAMPs[62], was not induced in GCs (Fig. 3E). In contrast, *LipoP1*, a highly inducible immune gene with a distinct expression profile from *FRK1*[6], was strongly expressed in GCs adjacent to infected pavement cells (Fig. 3F). Together, these findings reveal that SA biosynthesis and signaling are not components of GC-autonomous immunity, immune responses intrinsic to individual GCs and distinct from their role in regulating stomatal movement. This cell-type-specific immune program in GCs versus pavement cells represents a conserved pattern across infections by fungal (*C. higginsianum*) and bacterial (*P. syringae*) pathogens (Fig. 4, Supplementary Fig. 9). Single-cell RNA-seq data further corroborate this distinction, demonstrating that GCs and pavement cells possess unique transcriptional signature under both fungal and bacterial pathogen challenge (Fig. 4, Supplementary Fig. 10). Collectively, these observations suggest that although GCs can perceive pathogen attack, their intrinsic and cell-autonomous immune responses are selective and mechanistically distinct from those of pavement cells.

Plant defenses against fungal and bacterial pathogens share core signaling molecules, yet their spatiotemporal dynamics vary with the host, pathogen lifestyle, and specific molecular interactions. Upon perception of either bacterial or fungal challenges, rapid Ca$^{2+}$ influxes and an accompanying ROS burst arise in GCs and pavement cells, serving as early localized signals that initiate downstream immune responses[63,64]. Our findings show that GCs adjacent to *B. hordei*-infected pavement cells display a robust calcium influx (Fig. 5D, E) and pronounced ROS bursts (Fig. 5F, G), followed by cell autofluorescence indicative of hypersensitive cell death (Fig. 5H). Strikingly, ROS bursts in GCs preceded pathogen-induced SA biosynthesis in pavement cells during early infection (Supplementary Fig. 11A, B). Notably, ROS accumulation was also observed in GCs adjacent to an infection site of the adapted powdery mildew *E. cichoracearum* (Supplementary Fig. 11F–H). Guard cell death has been observed in several non-adapted rust fungus-plant interactions and consequently stops the fungal infection through the stomata[47,48]. Similar observations have been reported in interactions between wheat leaf rust fungus and non-host

Arabidopsis, where fungal penetration through stomata triggers transient stomatal closure followed by GC-specific hypersensitive cell death that prevents colonization[39,48]. It is known that GCs sense fungal chitin oligosaccharide (GlcNAc)$_8$ (CTOS) through its receptor CERK1 to induce calcium influx and ROS accumulation in GCs, which leads to activating Ca$^{2+}$- and anion channel SLAC1-dependent stomatal closure, whereas fungal conversion of CTOS to chitosan oligosaccharides (CSOS) suppresses this response and instead induces Ca$^{2+}$-dependent, CERK1-independent guard cell death[39], thereby rendering GCs functionally incompatible with host-adapted pathogens. These studies are consistent with our study, revealing that GCs engage a rapid and amplified immune signaling mode that is distinct from that of pavement cells.

In this study, we demonstrate that GC death can be triggered not only by direct infection with non-adapted powdery mildew but also by adapted powdery mildews. Key signaling molecules, calcium, ROS, and SA, coordinate the onset and progression of plant cell death in response to pathogen infection[65–67]. Using a transgenic line expressing *pAtML1::NahG-mSCARLET* to deplete SA accumulation specifically in epidermal cells, we show that *B. hordei*-induced cell death occurs independently of SA accumulation (Supplementary Fig. 12C–E). We propose that GCs exhibit a more robust calcium influx (Fig. 5D–E, Supplementary Fig. 11C–E) and pronounced ROS bursts (Fig. 5F, G, Supplementary Fig. 11A, F–H) compared with pavement cells, enabling them to execute hypersensitive cell death in response to pathogen infection without requiring SA signaling.

In monocots, each stoma comprises GCs accompanied by specialized subsidiary cells that support GC function. In barley *mlo* mutants, epidermal pavement cells exhibit strong resistance to barley powdery mildew, yet the subsidiary cells remain comparatively susceptible. Moreover, epidermal cells adjacent to invaded subsidiary cells also exhibit elevated susceptibility[68]. These observations indicate that distinct epidermal cell types in barley differ in their vulnerability to powdery mildew and that successfully colonized subsidiary cells can influence neighboring cells, rendering them more susceptible to

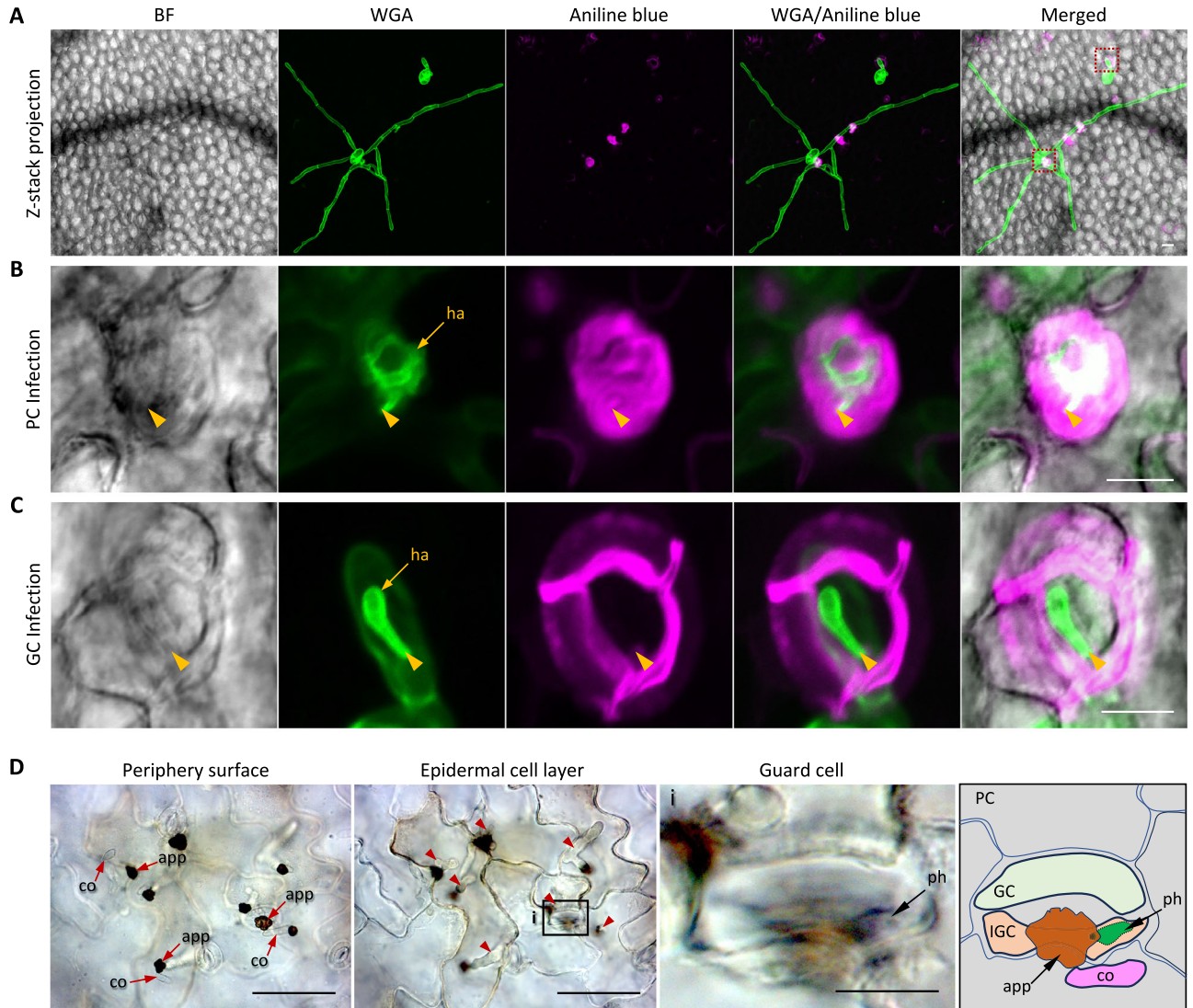

**Fig. 6 | Guard cells are incompatible with host-adapted pathogens. A–C** Confocal micrographs showing representative images of *Erysiphe cichoracearum* (*E. cichoracearum*) infection in a guard cell and a pavement cell of WT leaf tissue at 72 hpi. **A** Fungal development of *E. cichoracearum* following penetration into guard cells or pavement cells. Red dash frames mark the infected guard cell and pavement cell. Fungal structures were visualized using WGA (green), and callose deposition at infection sites was detected by aniline blue staining (magenta). Images are shown as Z-stack maximum intensity projections acquired by confocal microscopy. At least twenty single-infection sites were examined per cell type. Scale bar, 20 μm. **B** Enlarged single-plane image showing *E. cichoracearum* penetration in pavement cells from (**A**). Scale bar, 10 μm. **C** Enlarged single-plane image showing *E.* *cichoracearum* penetration in a guard cell from (**A**). Penetration sites are indicated by arrowheads; haustoria (ha) are marked with arrows. GC guard cell; PC pavement cell. Scale bar, 10 μm. **D** Representative light microscopy images showing *Colletotrichum higginsianum* (*C. higginsianum*) infection in pavement cells and guard cells of WT leaf tissue at 72 hpi. Red arrowheads indicate the development of primary hyphae in infected pavement cells. Scale bar, 50 μm. (i) Enlarged view showing detailed *C. higginsianum*-infection in a guard cell. At least twenty infected sites per cell type were examined. Scale bar, 10 μm. Schematic illustration (right panel) depicts the stomatal structures associated with *C. higginsianum* penetration. co conidiospore; app appressorium; ph primary hypha; IGC infected guard cell; PC pavement cell.

pathogen invasion. Unlike the powdery mildew fungi, *Zymoseptoria tritici*, the causal agent of Septoria tritici blotch in wheat, cannot directly penetrate the cuticle and instead enters the plant through stomata. Wheat cultivars with different types of resistance have been shown to block invasion by avirulent *Z. tritici* isolates during stomatal penetration[69,70]. Using both wheat-*Z. tritici* and wheat-barley powdery mildew (*B. hordei*) pathosystems, Valente et al. (2022) demonstrated that subsidiary cells in the wheat epidermis were the first to respond to *Z. tritici* infection by forming cell wall papillae enriched with callose at fungus contact sites. Similarly, *B. hordei* infection triggers papilla formation and callose deposition in subsidiary cells, although the callose deposition patterns differ between two different fungal pathogens[68,69]. Notably, no significant callose deposition is detectable in GCs[69]. Furthermore, none of these studies assessed the immune response or

immune status of the GCs themselves. Using Arabidopsis-powdery mildew pathosystems, our results are consistent with these observations, revealing a marked deficiency in callose deposition in GCs upon powdery mildew infection (Figs. 5A, B, 6A–C, Supplementary Fig. 13). Our systemic and comparative analyses of immune responses in GCs versus pavement cells will therefore be instrumental for advancing our understanding of epidermal cell-type-specific immunity. Nonetheless, stomatal architecture and mechanics vary substantially among plant species; a potential divergence in cell-type-specific immune mechanisms warrants further investigation.

In conclusion, our study provides a high-resolution, spatially explicit single-cell view of immune responses in Arabidopsis during powdery mildew infection, complemented by comparative imaging and transcriptomic analyses across diverse bacterial and fungal

pathosystems. We uncover striking differences between GCs and pavement cells in their activation of immune pathways. Although GCs are largely inactive in SA biosynthesis and related signaling, they remain resistant to pathogen colonization, implying the operation of alternative, SA-independent immune mechanisms. We propose that elucidating the molecular basis of GC-autonomous immunity will be pivotal for understanding spatial defense coordination in plants and could guide the engineering of durable disease resistance in crops.

## Method

### Plant materials and growth conditions

All *Arabidopsis thaliana* (Arabidopsis) lines used in this study were derived from the Columbia-0 (Col-0) ecotype. T-DNA insertion mutants included *sid2-1*[11], *cbp60g-1 sard1-2*[71] (CS72191) and *camta1/2/3*[26] (CS71606). Reporter lines utilized were *pICS1::NLS-3×mVENUS*[12] (CS2107988), *pPR1::NLS-3×mVENUS*[12] (CS2107991), *pHEL::NLS-3×mVENUS*[12] (CS2107982), *pLipoP1::NLS-3×mCITRINE*[6] (CS73266), *pFRK1::NLS-3×mVENUS*[6] (CS2110219), N7 nuclear marker (CS84731), PT-CK (CS16265), and MatryoshCaMP6s[34] (CS69957), all of which were obtained from the Arabidopsis Biological Resource Center (ABRC). The *CENH3-mRFP* line (N799456) was sourced from the Nottingham Arabidopsis Stock Center (NASC). The transgenic line *NahG* has been described previously[17]. DM398 plants, used for cloning the *mSCARLET-NOS TERMINATOR* cassette, were provided by Dr. Chris Ambrose. Transgenic lines generated in this study included *pICS1::ICS1-mSCARLET sid2-1* and *pEDS5::NLS-mSCARLET*. Seeds were germinated either in soil or on half-strength Murashige and Skoog (MS) medium and grown in a controlled environment chamber under a 16 h light/8 h dark photoperiod at 22 °C.

### Plasmid construction and plant transformation

To generate the *pICS1::ICS1-mSCARLET* and *pEDS5::NLS-mSCARLET* constructs, the *mSCARLET-NOS* DNA fragment was amplified from DM398 genomic DNA using primers Pac1-mScarlet-F and Spe1-mScarlet-R, and cloned into the pMDC123 vector[72] via enzyme restriction (NEB, Canada) and T4 DNA ligase-mediated ligation (Cat# EL0011, Thermo Fisher Scientific), resulting in the *mSCARLET-pMDC123* construct. For the *pICS1:ICS1-mSCARLET* construct, the *ICS1* genomic region (-2.1 kb upstream of the ATG to the end of the open reading frame excluding the stop codon)[12] was amplified from Col-0 genomic DNA using primers *Asc1-pICS1-F/Pac1-gICS1-R*, and inserted into the *mSCARLET-pMDC123* vector. For the *pEDS5::NLS-mSCARLET* construct, the *EDS5* promoter region (-1.5 kb upstream of the ATG)[18] was amplified using primers *Asc1-pEDS5-F* and *Pac1-NLS-pEDS5-R*, and ligated into *mSCARLET-pMDC123*. To generate the *pATML1::NahG-mSCARLET* construct, the NahG coding sequence was amplified from plasmid MT363 (Addgene plasmid#65858) using primers Asc1-NahG-FOR and Pac1-NahG-REV. The *ATML1* promoter region (3 kb upstream of the ATG) was amplified from Col-0 genomic DNA using primers Pme1-AtML1-FOR and Asc1-AtML1-REV. Both fragments were subsequently ligated into the *mSCARLET-pMDC123* vector.

Plant transformations were performed using the *Agrobacterium*-mediated floral dip method[73] to introduce the constructs into *sid2-1* or Col-0 backgrounds. Transgenic plants were selected using either hygromycin or Basta resistance, and positive lines were verified by confocal fluorescence imaging. At least ten independent transgenic lines were obtained and analyzed for each construct. Primer sequences are listed in Supplementary Table 1.

### Reciprocal crosses and plant genotyping

Flowers used for crosses at stage 12 were emasculated one day prior to hand pollination. Seeds resulting from these crosses were harvested and collected for downstream analyses. The genotypes of F1 progeny were confirmed by either PCR-based genotyping or confocal fluorescence imaging. Primer sequences used to confirm

T-DNA insertions in *CBP60g*, *SARD1*, *CAMTA1*, *CAMTA2*, and *CAMTA3* are provided in Supplementary Table 1. The *sid2-1* mutant allele carries a C to T substitution in exon 9[11], which creates an Mfe1 restriction site. PCR fragments amplified using genotyping primers listed in Supplementary Table 1 were digested with Mfe1 to confirm the mutation. The presence of the *NahG* transgene in *NahG pICS1::NLS-3 × mVENUS pEDS5::NLS-mSCARLET* plants was verified by PCR-based genotyping using *NahG*-specific primers (Supplementary Table 1).

### Pathogen maintenance and plant inoculation

Powdery mildew strains were maintained as described previously[14]. Briefly, the barley powdery mildew *Blumeria hordei* (*B. hordei*) and cucumber powdery mildew *Erysiphe cichoracearum* (*E. cichoracearum*) were maintained on barley (*Hordeum vulgare* L, cultivar CDC silky) and cucumber (*Cucumis sativus*, variety Sweet Slice, McKenzie, Canada) plants, respectively. The canola powdery mildew *Erysiphe cruciferarum* (*E. cruciferarum*) was propagated on canola (*Brassica napus* L. *cv.* DH12075). Arabidopsis plants were placed in settling towers and inoculated with conidia by tapping heavily infected barley, cucumber or canola leaves above the Arabidopsis plants. After 30 min, the inoculated plants were returned to the growth chamber for further investigation. Low-density inoculations (5 to 10 conidia mm⁻²) were employed for this study[17].

The anthracnose pathogen *Colletotrichum higginsianum* (*C. higginsianum*) was cultured and handled as described previously[17,32]. Arabidopsis plants were sprayed with conidial suspensions on the leaf surface. Immediately after inoculation, the inoculated plants were placed in a 100% humidity chamber for 48 h.

*Pseudomonas syringae* pv tomato DC3000 strains harboring plasmid pVSP61 (virulent strain) or pV288 containing the *avrRpt2* gene (avirulent strain) and secretion protein defective strains (*hrc C*, *hrp A* and *hrp S*) were grown at 29 °C in liquid King's B medium. Overnight log phase cultures were grown to an optical density at $OD_{600}$ of ~0.6 and diluted with sterile water before inoculation. The bacterial suspensions were infiltrated into the abaxial surface of Arabidopsis leaves using a 1-mL syringe without a needle. Control inoculations were performed with sterile water[17].

### Leaf tissue fixation

The leaf tissues were fixed in 1 ml of methanol:chloroform:acetic acid (6:3:1, v/v/v) solution overnight at room temperature. The samples were then rehydrated through a graded ethanol series (30 min per step): 100%, 90%, 80%, 70%, 50%, 30%, 10%, and distilled water. After rehydration, the leaf samples were stored in 150 mM $K_2HPO_4$ buffer (pH 9.5) at 4 °C until further use.

### Chemical treatment of plants

Lanthanum chloride ($LaCl_3$), ruthenium red, verapamil, DNQX, W-7, erythrosin B, eosin Y, and CPA were purchased from Sigma-Aldrich. One hour prior to *B. hordei* inoculation, Arabidopsis leaves were infiltrated with 5 mM $LaCl_3$[24,74], 1 mM ruthenium red[24,75,76], 10 mM verapamil[24,77], 0.5 mM DNQX[24,78,79], 50 μM W-7[24,80], 1 μM erythrosin B[24,81], 0.5 μM eosin Y[24,81], or 25 μM CPA[24,81]. Plants infiltrated with sterile water served as mock controls. Leaf samples treated with either $Ca^{2+}$ blocker or water were examined at 24 h post inoculation (hpi) using confocal laser scanning microscopy (ZEISS LSM 880).

For Calcium-deficiency treatments, plants were grown hydroponically as described previously[82], with minor modifications. Briefly, Arabidopsis seeds were surface-sterilized and sown onto 0.7% (w/v) agar, aliquoted into 10-μl pre-cut pipette tips floating on hydroponic solution containing 0.5 mM $Ca^{2+}$. After one week of growth, seedlings were transferred to fresh hydroponic solutions with varying $Ca^{2+}$ concentrations: 0.025 mM, 0.1 mM, and 0.5 mM. At three weeks of age, plants grown under different calcium concentrations were inoculated

with *B. hordei* and analyzed at 24 hpi using the ZEISS LSM 880 confocal microscope.

## Microscopy and imaging analysis

Fixed leaf tissues were stained with 0.05% aniline blue and/or 10 µM Alexa Fluor 488-conjugated wheat germ agglutinin (WGA, Life Technologies). Stained samples were imaged using a ZEISS LSM 880 or Leica Stellaris 5 confocal laser scanning microscope. For fresh tissue imaging, leaf samples were incubated in either distilled water or 10 µM propidium iodide (PI, Life Technologies) prior to imaging. To visualize reactive oxygen species (ROS) accumulation in guard cells, leaf tissues, with or without powdery mildew inoculation, were vacuum-infiltrated with 20 µM 5-(and-6)-chloromethyl-2′,7′-dichlorodihydrofluorescein diacetate (CM-H$_2$DCFDA) (Invitrogen, Oregon, USA) for 15 min and immediately imaged by confocal microscopy. Macroscopic detection of H$_2$O$_2$ accumulation by 3,3′-Diaminobenzidine (DAB) staining was performed as described earlier[32]. To visualize fungal structures, *C. higginsianum*-infected Arabidopsis leaf tissues were stained with trypan blue[17,32].

Fluorescence signals was detected using the following excitation/emission wavelength settings: CFP and aniline blue (405/420–480 nm), GFP and WGA (488/505–530 nm), CM-H$_2$DCFDA (488/517–527 nm) mVENUS/mCITRINE (514/530–560 nm), mORANGE (543/550–570 nm), mRFP and mSCARLET (543/581–635 nm), PI (488/610–630), and chlorophyll A (594/ > 650 nm). All images were processed and analyzed using LSM Image Browser (Zeiss) and Fiji/ImageJ (2.14.0) software.

## Measurement of nuclear and cytoplasmic signal intensity

Nuclear and cytoplasmic signal intensity measurements were conducted using ImageJ (version: 2.14.0). Confocal images were first converted to 8-bit greyscale, and regions of interest (ROIs) corresponding to individual nuclei or cell cytoplasm were outlined based on fluorescent reporter signals. For each ROI, the mean gray value was measured to quantify fluorescence intensity. Background fluorescence was determined from an adjacent non-fluorescent area and subtracted from each measurement to obtain the corrected nuclear or cytoplasmic signal intensity. All measurements were performed using identical acquisition settings to ensure comparability across samples. The number of biological replicates analyzed in each experiment is indicated in the corresponding figure legends.

## Single-cell RNA-seq data analysis

The preprocessed single-cell RNA-seq datasets were obtained from the INRAE dataverse repository and analyzed following the published pipeline[34]. Briefly, data processing and clustering were performed using the Seurat R package[83]. Genes expressed in three cells or fewer cells and cells expressing fewer than 200 genes were excluded from the analysis. Only cells containing ≤5% mitochondrial and chloroplastic reads were retained for downstream analyses. Data normalization was conducted using the SCTransform function, and principal component analysis (PCA) was performed on the 2000 most variable genes identified with the FindVariableFeatures function, retaining 30 principal components. Cell clustering was carried out using the FindClusters function with a resolution parameter of $r = 1$. Cell-type identities were assigned based on known marker genes[34]. Transcriptional dynamics were analyzed using the Monocle3 package[84], and individual gene expression patterns across cell clusters were visualized with the plot_cells function in Monocle3.

## GO analysis

GO analysis for enriched biological processes was conducted with an online tool (http://geneontology.org/). GO terms were identified with an indication of "overrepresentation" by fold enrichment and *p*-values.

## Statistical analysis

All statistical analyses were performed using R software (v.4.1.2) (http://r-project.org). Comparisons between two groups were conducted using a two-tailed Student's *t*-test, whereas comparisons among more than two groups were performed using one-way ANOVA followed by Tukey's multiple comparisons test. Boxplots were generated in RStudio using appropriate data visualization packages.

## Reporting summary

Further information on research design is available in the Nature Portfolio Reporting Summary linked to this article.

## Data availability

All constructs and transgenic lines generated in this study are available from the corresponding author upon request. Source data are provided with this paper.

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

## Acknowledgments

We thank ABRC and NASC for providing the mutant seeds and reporter lines used in this study. We also thank Dr. Chris Ambrose for providing the DM398 seeds. This work was supported by grants from the Natural Science and Engineering Research Council of Canada (Discovery Grant) and the Diverse Field Crop Cluster: Agriculture and Agri-Food Canada to Y.W.

## Author contributions

Y.W. and J.S. conceived and coordinated the study; Y.W. and J.S. designed the experiments. J.S. and M.M. conducted plant transformation and transgenic line screening. D.K. and W.A. maintained powdery mildews and performed pathogen inoculation. J.S. conducted microscopy and imaging; J.S., M.M., and Y.W. conducted data analyses. J.S. and Y.W. wrote the manuscript. J.S., Y.C., and Y.W. revised the manuscript and were involved in the discussion of the work.

## Competing interests

The authors declare no competing interests.
