## [Transparent Peer Review file · Nature Communications]

Cell-type-specific immune programs orchestrate spatial defense in the Arabidopsis leaf epidermis

Corresponding Author: Dr Yangdou Wei

Version 0:

Reviewer comments:

Reviewer #1

(Remarks to the Author)

The manuscript by Song and co-workers reports cell-type-resolved analyses of powdery mildew attack in *Arabidopsis thaliana*. The authors used mainly transgenic reporters, but also histochemical staining, to illuminate the type of defence responses in epidermal pavement cells as opposed to epidermal guard cells. They employed both adapted and non-adapted powdery mildew species in this context. They found that stomatal guard cells do not trigger a salicylic acid-dependent immune response to powdery mildew fungi, suggesting cell-type-specific differences in the activation of immunity. Guard cells were further found to be resistant to adapted powdery mildew fungi, undergoing early hypersensitive cell death upon host cell invasion.

I consider the topic of cell-type-specific immune responses highly interesting. As such, the theme of the present study is of high biological significance. The technical quality of the work performed is high (though I would have welcomed an even more quantitative assessment of the reported cellular features and infection phenotypes), and the manuscript is well written. Some of the micrographs are of spectacular quality. I also appreciate the technical challenges likely associated with the work performed. However, unfortunately, the study remains largely descriptive, reporting distinctive cellular activities in pavement and guard cells in response to adapted or non-adapted powdery mildew pathogens. It is, of course, necessary to perform such careful observational work as a first step for further analysis. For a journal such as *Nature Communications*, I would have expected, however, that the authors go one step further to unravel the mechanistic basis for the reported differences. One possibility would be to perform single-cell RNA sequencing, as recently published in the context of a different fungal pathogen ([https://www.cell.com/cell-host-microbe/pdf/S1931-3128\(23\)00344-X.pdf](https://www.cell.com/cell-host-microbe/pdf/S1931-3128(23)00344-X.pdf)). Alternatively or in addition, cell type-specific manipulation of plant immunity (see <https://academic.oup.com/plcell/article/31/12/2868/5985837>) could have provided additional insights.

Specific comments

Line 1: Title: The title is too general (“pathogenic fungi”) as only interactions with powdery mildew fungi were looked at.

Line 96: The taxonomy of the barley powdery mildew pathogen has been recently revised, and the fungus renamed “*Blumeria hordei*” (<https://pmc.ncbi.nlm.nih.gov/articles/PMC9157761/>).

Lines 130-133: I do not think the conclusion here is complete/correct. An alternative interpretation is that the adapted powdery mildew pathogen is capable of suppressing ICS1 induction/cell death in neighboring cells.

Lines 153-155: I do not think the conclusion here is complete/correct. There is no direct evidence that provided that cell death amplifies SA biosynthesis.

Lines 198-202: I am surprised that the authors employed the NahG transgene for these experiments. Why didn't the authors make use of the *sid2* mutant? Was the expression of the NahG gene checked in the progeny of the crosses? NahG gene expression comes with the undesired accumulation of catechol.

Line 260: It is actually sirofluor, not aniline blue, which stains callose (<https://pubmed.ncbi.nlm.nih.gov/38739552/>).

Line 293: Heading is too general – only powdery mildew fungi were tested.

Line 294: “showed” should read “shown”

Lines 359-360: Isn't it surprising that guard cells, despite the activation of fewer immune pathways (as far as one can tell from the present study), have a better immune status? The authors gloss over this conundrum, which should be mentioned/discussed.

Line 366 ff: It would have been valuable if the authors, besides the broad-spectrum calcium channel blocker lanthanum

chloride, also used more selective calcium channel blockers that inhibit members of individual calcium channel families.

The Discussion omits important points. How do the authors rationalize their findings in the context of a general SA-responsiveness of guard cells (e.g., <https://pmc.ncbi.nlm.nih.gov/articles/PMC6130018/>, <https://pmc.ncbi.nlm.nih.gov/articles/PMC6512940/>, and many other publications)? Further, findings of guard cells in the context of immunity against filamentous pathogens are not properly integrated into the Discussion (e.g., <https://www.pnas.org/doi/10.1073/pnas.1922319117> and <https://link.springer.com/content/pdf/10.1186/s12870-024-05426-5.pdf>). It is also known that subsidiary cells of the stomatal complex can exhibit aberrant immunity to powdery mildew fungi. In barley mlo mutants, which are otherwise highly resistant against the barley powdery mildew pathogen, subsidiary cells remain susceptible (see https://www.apsnet.org/publications/phytopathology/backissues/Documents/1977Articles/Phyto67n05_678.PDF). The discrepancy between powdery mildew-resistant guard cells and powdery mildew-susceptible subsidiary cells would be worthwhile to discuss. A special role of subsidiary cells is also highlighted in the following work: <https://www.biorxiv.org/content/10.1101/2022.07.15.499463v1>.

Reviewer #2

(Remarks to the Author)

Title: Deciphering Guard Cell-Autonomous Immunity Against Pathogenic Fungi (NCOMMS-25-57522)

In this study, authors used transgenic immune reporter systems and live imaging to analyze cell-type specific immune responses in Arabidopsis leaves infected with different non-adapted and adapted pathogenic fungi. Study showed that while neighboring epidermal cells activated salicylic acid signaling and callose deposition, guard cells showed distinct immunity characterized by strong calcium influx, reactive oxygen species bursts, and hypersensitive cell death. Although the findings of the study are indeed novel and will advance our understanding of the autonomous and unique defense mechanisms occurring in guard cells, the actual mechanism of GC-mediated defense response is not clear even though the authors showed Ca signaling and ROS are involved. In addition, the authors need to address the following issues regarding some technical aspects and the claims made.

- How many transgenic lines exhibited similar expression patterns for pICS1::ICS1-mSCARLET and pEDS5::NLS-mSCARLET? Since these lines have not been reported before, please present data from at least 2–3 independent lines showing consistent expression patterns in response to fungal pathogens. These data can be included as supplementary material.
- This study used several mutants (e.g., eds5, sard1, cbp60g, camta123, etc.) related to different defense-related pathways, which were crossed with reporter lines. Although the primer sequences used for genotyping are provided in the supplementary material, no data confirming the genotyping of these lines for homozygosity and knockout status of the target genes are included. Please provide these data as supplementary material.
- Although the study analyzed different fungi, terms such as 'pathogen attack,' 'pathogen-induced SA,' or 'pathogen-induced' such as in lines 53 and 57, are used throughout the manuscript when reporting observations in response to fungal infection. Please be specific to fungal pathogens, as responses to other pathogens (bacterial or viral) were not tested and may differ.
- How would you explain a faint cyan color in the neighboring cells for adapted fungus (supplemental Fig. 1A)?
- In Fig. 2D, the pICS1-driven reporter construct shows low activity; therefore, the claim in lines 187–188 that NLS-3xmVENUS signals are completely absent in the cbp60g sard1 mutants does not appear to be accurate. Additionally, there is no evident saturation of pICS1::NLS-3xmVENUS signals in the camta1/2/3 mutant upon B.g. hordei infection. In fact, the signal appears weaker in B.g. hordei-infected tissue (Fig. 2D) compared to the non-infected control (Fig. 2C). Why is signal quantification provided for the reporter in non-infected tissues in Supplemental Fig. 4B (based on Fig. 2C), but not for the infected tissues shown in Supplemental Fig. 4 (based on Fig. 2D)?
- Line 217-188, the authors mention examining the reporter gene expression in infected GCs. How often do they find this? Some numbers or frequency will help.
- Lines 267–268: There appears to be callose deposition in one of the guard cells (GCs) of the kidney-shaped structure in Fig. 4B and C. Although it seems slightly away from the penetration site, it still appears to be part of the GC. A similar pattern of callose deposition is observed in the GCs in Supplemental Fig. 7B for the adapted fungal pathogen. Therefore, it is unclear from these images whether the callose was deposited by the GCs or by neighboring pavement cells, as callose is always deposited in the apoplast (outside the cell membrane).
- The GC-mediated defense against adapted pathogen is intriguing but the mechanism is not shown. For non-adapted pathogen, the authors show Ca signaling ROS play a role in cell defense. Is it the same for adapted pathogens? This can be easily tested. This observation also warrants more discussion. Why an adapted pathogen not able to suppress defense response in GCs? Does GCs lack components needed for fungal cell differentiation and development?
- Discussion section is very poorly written. It is just a summary of results and repetitive rather than discussion with reference to previous work. There are numerous studies regarding the role of GCs in plant defense. Surprisingly, only three additional references are added in the discussion! In addition, the authors could hypothesize or come up with a model regarding how the GC-mediated defense response is triggered? For example, is the defense response triggered directly by PAMPs or internal signal from neighboring cells?

Minor Comments

- Figure legend 4A. is it really a “cyan color or magenta” representing aniline blue staining?
- Also include a paragraph on comparing cell-type specific defense responses (Ca²⁺ signaling, ROS accumulation, SA production) between fungal and bacterial pathogens in the discussion section.
- Line 108: “emerge: instead of “merge”?

- Line 213: replace GSs (GCs)
- Line 282: Expand or explain H2DCF-DA. Readers may not be familiar with this.
- Line 294: replace “non-adapted fugus pathogens” with “non-adapted fungal pathogens”.
- Throughout the manuscript either use *A. thaliana* or *Arabidopsis* (common name with no italics).
- The title of the manuscript can be a bit more descriptive and interesting to describe the actual findings of the manuscript.

Reviewer #3

(Remarks to the Author)

Version 1:

Reviewer comments:

Reviewer #1

(Remarks to the Author)

The resubmitted version of the manuscript by Song and co-workers shows impressive improvements. The authors addressed all my concerns and now provide several new experimental data and in silico analyses of single-cell RNA-Seq data. Major improvements cover the following aspects:

- 1.) Marker fluorescence intensities were quantified. This approach further strengthens the claims of the authors.
- 2.) Pre-existing single cell RNA-Seq data from infection experiments (conducted by other groups) with *C. higginsianum* and *P. syringae* were analyzed. Careful examination of these data sets supports the key conclusions of the present manuscript. Using pre-existing published single-cell transcriptomic data is a creative approach to address this issue. Supplementing this, infection experiments with ICS1 reporter lines were performed to examine for a putative cell type-specific activation of the SA signaling pathway in these interactions.
- 3.) In addition to the NahG transgene, the *sid2* mutant was used for crossings with the ICS1 reporter line. The results of both approaches are consistent. I do appreciate the rationale of the authors to keep the NahG transgene, as this shows a stronger effect.
- 4.) A panel of more specific calcium inhibitors has been tested (at single concentrations only, though), which revealed a key role for P-type calcium ATPases in the activation of pathogen-induced ICS1 expression. Maybe a short note could be added to justify the concentrations used.
- 5.) The Discussion was largely expanded and improved. It covers many more aspects, placing the findings in a broader context. It now has sufficient breadth and depth.

In addition to the key issues, the authors also dealt with the minor points mentioned.

I have the impression that also the concerns of reviewer #2 were addressed adequately to the most part, as we shared some similar comments, e.g., regarding the lack of more mechanistic insights.

In summary, I feel that the current version has improved substantially. The authors have made a major and serious effort to advance their study. In my opinion, the work has reached the level required for publication in a journal like Nature Communications. Unraveling cell type specificity in immune responses is a topic of broad scientific relevance. I congratulate the authors on an important advancement in our understanding of plant immunity.

Reviewer #2

(Remarks to the Author)

The authors have substantially improved the manuscript by adding a new dataset that supports their findings on distinct defense responses in pavement and guard cells, namely salicylic acid signalling and callose deposition in pavement cells and strong Ca²⁺ influx, amplified ROS production, and SA-independent cell death in guard cells. Furthermore, the conservation of these cell-type-specific responses to both bacterial and fungal infections make the study particularly interesting and novel. Additionally, the authors have successfully addressed the issues raised in the previous submission

Reviewer #3

(Remarks to the Author)

Response to Reviewers' Comments

We are grateful to the editor for providing early notice of the reviewers' comments, which afforded us ample time to conduct additional genetic, molecular, cellular, and pathological assays. This additional work has allowed us to fully address the reviewers' suggestions and further improve the manuscript. Detailed responses to each specific comment are provided below.

Reviewer's Comments:

Reviewer #1 (Remarks to the Author)

The manuscript by Song and co-workers reports cell-type-resolved analyses of powdery mildew attack in Arabidopsis thaliana. The authors used mainly transgenic reporters, but also histochemical staining, to illuminate the type of defence responses in epidermal pavement cells as opposed to epidermal guard cells. They employed both adapted and non-adapted powdery mildew species in this context. They found that stomatal guard cells do not trigger a salicylic acid-dependent immune response to powdery mildew fungi, suggesting cell-type-specific differences in the activation of immunity. Guard cells were further found to be resistant to adapted powdery mildew fungi, undergoing early hypersensitive cell death upon host cell invasion.

1. I consider the topic of cell-type-specific immune responses highly interesting. As such, the theme of the present study is of high biological significance. The technical quality of the work performed is high (though I would have welcomed an even more quantitative assessment of the reported cellular features and infection phenotypes), and the manuscript is well written. Some of the micrographs are of spectacular quality.

Response: We appreciate the reviewer's thoughtful assessments of the significance and quality of our work. Regarding the quantitative assessment of our data, we have incorporated detailed measurements of fluorescence signal intensities in the revised manuscript. Specifically, we conducted a comprehensive analysis of nuclear and cytoplasmic *NLS-3×mVENUS* signal intensities to quantify the temporal induction of *ICS1* transcription during *B. hordei* powdery mildew infection (see revised Fig. 1A-D). In the revised manuscript, we now state:

*“For quantitative analysis of signal induction, fluorescence intensities were measured in both the nuclei and the cytoplasm of epidermal cells either attempted or penetrated by *B. hordei* (designated as “Patient 0”, P), as well as in their successive neighboring epidermal cell layers. Cells in direct contact with “Patient 0” were defined as the first layer (L1), while cells adjacent to L1, but not directly bordering “Patient 0”, constitute the second layer (L2), and so forth (Supplemental Fig. 1A). At 18 hpi, no detectable induction of *NLS-3×mVENUS* fluorescence was observed in either the nuclei or the cytoplasm of cells at the penetration site (Fig. 1A-B). By 21 hpi, however, visible induction of nuclear-associated fluorescent signals became apparent in “Patient 0” and adjacent L1 cells (Fig. 1A-B)” (lines 108-117).*

“At 24 hpi, a strong NLS-3×mVENUS fluorescence signal was observed in “Patient 0” cells exhibiting visible haustorial formation (Fig. 1C-D). Notably, fluorescence induction was also detected in adjacent L1 and L2 cells, forming a prominent patchy induction pattern in which nuclear signal intensity peaked at the center and gradually declined toward the periphery (Fig. 1C-D). In the central region of the patch, fluorescence was evident not only in the nuclei but also in the cytoplasm of L1 and L2 cells (Fig. 1C-D).” (lines 122-127).

Additionally, we carried out intensity analysis of nuclear and cytoplasmic NLS-3×mVENUS signals in *E. cichoracearum*-infected cells (see revised Supplemental Fig. 1B-D). The revised manuscript now provides the following description:

“At 26 hpi, induction of both nuclear and cytoplasmic NLS-3×mVENUS signals was evident in *E. cichoracearum*-penetrated cells and their neighboring cells, resembling the ICSI induction pattern observed during *B. hordei* infection (Supplemental Fig. 1B-D). These findings indicate that ICSI expression is triggered in response to infection by both adapted and non-adapted powdery mildew fungi.” (lines 132-137).

2.I also appreciate the technical challenges likely associated with the work performed. However, unfortunately, the study remains largely descriptive, reporting distinctive cellular activities in pavement and guard cells in response to adapted or non-adapted powdery mildew pathogens. It is, of course, necessary to perform such careful observational work as a first step for further analysis. For a journal such as *Nature Communications*, I would have expected, however, that the authors go one step further to unravel the mechanistic basis for the reported differences. One possibility would be to perform single-cell RNA sequencing, as recently published in the context of a different fungal pathogen ([https://www.cell.com/cell-host-microbe/pdf/S1931-3128\(23\)00344-X.pdf](https://www.cell.com/cell-host-microbe/pdf/S1931-3128(23)00344-X.pdf)). Alternatively or in addition, cell type-specific manipulation of plant immunity (see <https://academic.oup.com/plcell/article/31/12/2868/5985837>) could have provided additional insights.

Response: We appreciate the reviewer’s constructive comments and suggestions, which underscored the importance of examining the mechanistic basis underlying the differences between pavement cells and guard cells, including the recommendation to perform single-cell sequencing analysis to address this question.

To address these points, we expanded our analyses in the following ways:

First, we tested an additional fungal pathogen, *Colletotrichum higginsianum* (*C. higginsianum*), using our reporter line *pICS1::NLS-3×VENUS CENH3-mRFP* (Fig. 4A). At 48 hours post-inoculation (hpi), strong NLS-3×VENUS fluorescence was detected in *C. higginsianum*-penetrated epidermal cells containing visible primary hyphae. Fluorescence was also observed in surrounding cells, forming a distinct patchy induction pattern with the highest nuclear signal intensity at the center, gradually decreasing toward the periphery. Importantly, *ICSI* transcriptional activation was not detected in guard cells, consistent with our observations in powdery mildew infection. To further investigate the transcriptional differences between cell types

during fungal infection, we re-analyzed single-cell transcriptome data generated by Tang et al. (2023). Although the dataset covered multiple cell types, we specifically focused on guard cells and pavement cells at *C. higginsianum* infection sites. Our analysis revealed that pavement cells and guard cells exhibit distinct gene induction profiles (Fig. 4B). GO term analysis indicated that these two cell types undergo different biological processes during pathogen attack (Fig. 4C). Notably, the chorismate biosynthesis pathway was specifically induced in infected pavement cells but not in guard cells. In Arabidopsis, pathogen-induced salicylic acid (SA) synthesis depends on the conversion of chorismate to isochorismate via ICS1. These findings suggest that distinct transcriptional programs in pavement cells versus guard cells lead to divergent pathogen-response pathways.

To extend our study beyond fungal pathogens, we also examined Arabidopsis plants expressing *pICS1::NLS-3×mVENUS* and *pICS1::ICS1-mSCARLET* during infections with various bacterial *Pseudomonas syringae* pv. *tomato* (*Pst*) DC3000 strains. Similar to our findings with fungal pathogens, *Pst* infection triggered robust *ICS1* transcription and translation in pavement cells but not in guard cells (Fig. 4D-E, Supplemental Fig. 9). This indicates that the *ICS1* induction pattern we observed is conserved across both fungal and bacterial infections. Several independent groups have recently reported cell-type-specific responses to *Pst* at single-nucleus resolution (Nobri et al., 2025; Delannoy et al., 2023; Zhu et al., 2023). Among these, Delannoy et al. provided a comprehensive analysis of cell specialization and coordination during *Pst* challenge. We reanalyzed the datasets generated by Delannoy et al. and compared the induced GO terms between pavement and guard cells (Fig. 4F). Consistent with our findings, SA signaling was activated only in pavement cells, not in guard cells. Moreover, pathogen-induced SA biosynthetic genes, including *ICS1* and *PBS3*, were strongly upregulated in pavement cells but absent in guard cells (Fig. 4G).

Together, these new experimental results and our reanalyses of publicly available single-cell transcriptomic datasets reveal a conserved and fundamental distinction between pavement cells and guard cells in their transcriptional responses to pathogen attack. These findings directly address the key concerns raised by the Reviewers and further strengthen the conclusions of our manuscript. The corresponding methodological and result descriptions have been incorporated into the revised version as indicated.

Specific comments

3. Line 1: Title: The title is too general (“pathogenic fungi”) as only interactions with powdery mildew fungi were looked at.

Response: In this revised manuscript, we have extended our analysis to additional pathogen types and have updated the title to “*Cell-type-specific immune programs orchestrate spatial defense in the Arabidopsis leaf epidermis*”.

4. Line 96: The taxonomy of the barley powdery mildew pathogen has been recently revised, and the fungus renamed “*Blumeria hordei*” (<https://pmc.ncbi.nlm.nih.gov/articles/PMC9157761/>).

Response: We thank the reviewer for this updated information. The pathogen name has been revised to “*Blumeria hordei* (*B. hordei*)” throughout the manuscript.

5. Lines 130-133: I do not think the conclusion here is complete/correct. An alternative interpretation is that the adapted powdery mildew pathogen is capable of suppressing ICS1 induction/cell death in neighboring cells.

Response: We carefully re-examined the interactions between the adapted powdery mildew pathogen *E. cichoracearum* and *Arabidopsis*. Following *E. cichoracearum* infection at 26 hpi, we observed the patchy induction pattern of NLS-3×mVENUS in the ICS1 reporter line, similar to that observed with *B. hordei* infection. The revised manuscript has been updated accordingly.

“At 26 hpi, induction of both nuclear and cytoplasmic NLS-3×mVENUS signals was evident in *E. cichoracearum*-penetrated cells and their neighboring cells, resembling the ICS1 induction pattern observed during *B. hordei* infection (Supplemental Fig. 1B-D). These findings indicate that ICS1 expression is triggered in response to infection by both adapted and non-adapted powdery mildew fungi.” (lines 132-137).

6. Lines 153-155: I do not think the conclusion here is complete./correct. There is no direct evidence that provided that cell death amplifies SA biosynthesis.

Response: We thank the reviewer for the insightful comments. The observation of a patchy NLS-3×mVENUS induction pattern in the ICS1 reporter line upon a compatible *E. cichoracearum* infection suggests that cell death is not necessarily required to amplify SA biosynthesis. The revised manuscript has been updated accordingly.

7. Lines 198-202: I am surprised that the authors employed the *NahG* transgene for these experiments. Why didn't the authors make use of the *sid2* mutant? Was the expression of the *NahG* gene checked in the progeny of the crosses? *NahG* gene expression comes with the undesired accumulation of catechol.

Response: Following the reviewer's suggestions, we introduced the ICS1 reporter transgene, *pICS1::NLS-3×mVENUS*, into the *sid2-1* mutant background. The new data show that, following *B. hordei* infection at 24 hpi, ICS1 reporter expression was induced similarly to that observed in the WT background (Supplemental Fig. 6A-B).

In *Arabidopsis*, the ICS pathway is responsible for approximately 90% of SA production under stress conditions (Garcion et al. 2008, <https://doi.org/10.1104/pp.108.119420>). Therefore, the absence of *ICS1* does not completely abolish SA production. By contrast, overexpression of *NahG*, which converts SA to catechol, prevents SA accumulation and the establishment of SAR in *Arabidopsis* (Gaffney et al. 1993. DOI: 10.1126/science.261.5122.754). For this reason, *NahG* was included in our experiments. The presence of *NahG* in the experimental materials was confirmed

by PCR-based genotyping (Supplemental Fig. 7B). The revised manuscript has been updated with the following text:

“To further explore this regulatory network, we introduced the pICS1::NLS-3×mVENUS reporter into the ics1 mutant (sid2-1) and generated dual pICS1::NLS-3×mVENUS pEDS5::NLS-mSCARLET reporters in the SA-deficient NahG background through genetic crossing. Notably, following B. hordei infection, the reporters displayed comparable induction patterns in WT, sid2-1, and NahG backgrounds (Supplementary Fig. 6-7). These results suggest that B. hordei-induced activation of ICS1 and EDS5 occurs independently of SA or its precursor chorismite accumulation.” (lines 208-214)

8. Line 260: It is actually sirofluor, not aniline blue, which stains callose (<https://pubmed.ncbi.nlm.nih.gov/38739552/>).

Response: We thank the reviewer for the clarification and have updated the information accordingly in the revised manuscript (line 324).

9. Line 293: Heading is too general – only powdery mildew fungi were tested.

Response: For further clarification, in addition to two adapted powdery mildew species (*E. cichoracearum* and *E. cruciferum*), we also examined the distinct hemibiotrophic fungal pathogen *C. higginsianum* in our pathogenicity assays conducted on pavement and guard cells. In the manuscript, we state:

“We further tested GC susceptibility to C. higginsianum. In pavement cells, infection proceeded through appressoria, forming primary hyphae by 3 dpi (Fig. 6D, Supplemental Video 7), which subsequently developed into thin secondary hyphae by 4 dpi (Supplementary Fig. 13). In contrast, C. higginsianum infection in GCs was limited to a short initial hypha, which failed to progress (Fig. 6D, Supplemental Video 7).” (lines 404-409)

10. Line 294: “showed” should read “shown”.

Response: It is now corrected in the revised manuscript.

11. Lines 359-360: Isn't it surprising that guard cells, despite the activation of fewer immune pathways (as far as one can tell from the present study), have a better immune status? The authors gloss over this conundrum, which should be mentioned/discussed.

Response: We appreciate the reviewer's insightful comments. In addition to expanding the Discussion on the immune status of guard cells, we conducted several experiments to further elucidate the molecular mechanisms underlying their enhanced immune status compared with pavement cells. Our results show that, upon both adapted and non-adapted powdery mildew infection, guard cells activate a rapid and amplified immune response characterized by a pronounced Ca²⁺ influx and ROS burst, which is distinct from the response observed in pavement cells (Supplementary Fig. 11). We also found that powdery mildew-induced cell death occurs

independently of SA accumulation and signaling (*Supplementary Fig. 12*). In the revised manuscript, we state:

“Plant defenses against fungal and bacterial pathogens share core signaling molecules, yet their spatiotemporal dynamics vary with the host, pathogen lifestyle, and specific molecular interactions. Upon perception of either bacterial or fungal challenges, rapid Ca²⁺ influxes and an accompanying ROS burst arise in GCs and pavement cells, serving as early localized signals that initiate downstream immune responses” (lines 495-499)

“Guard cell death has been observed in several non-adapted rust fungus-plant interactions and consequently stops the fungal infection through the stomata. Similar observations have been reported in interactions between wheat leaf rust fungus and non-host Arabidopsis, where fungal penetration through stomata triggers transient stomatal closure followed by GC-specific hypersensitive cell death that prevents colonization. It is known that that GCs sense fungal chitin oligosaccharide (GlcNAc)₈ (CTOS) through its receptor CERK1 to induce calcium influx and ROS accumulation in GCs, which leads to activate Ca²⁺- and anion channel SLAC1-dependent stomatal closure, whereas fungal conversion of CTOS to chitosan oligosaccharides (CSOS) suppresses this response and instead induces Ca²⁺-dependent, CERK1-independent guard cell death, thereby rendering GCs functionally incompatible with host-adapted pathogens. These studies are consistent with our study revealing that GCs engage a rapid and amplified immune signaling mode that is distinct from that of pavement cells. (lines 505-516)

*In this study, we demonstrate that GC death can be triggered not only by direct infection with non-adapted powdery mildew but also by adapted powdery mildews. Key signaling molecules, calcium, ROS, and SA, coordinate the onset and progression of plant cell death in response to pathogen infection. Using transgenic line expressing pAtML1::NahG-mSCARLET to deplete SA accumulation specifically in epidermal cells, we show that *B. hordei*-induced cell death occurs independently of SA accumulation (*Supplemental Fig. 12C–E*). We propose that GCs exhibit a more robust calcium influx (*Fig. 5D–E, Supplemental Fig. 11C–E*) and pronounced ROS bursts (*Fig. 5F–G, Supplemental Fig. 11A, F–H*) compared with pavement cells, enabling them to execute hypersensitive cell death in response to pathogen infection without requiring SA signaling.”* (lines 517-525).

12. Line 366 ff: It would have been valuable if the authors, besides the broad-spectrum calcium channel blocker lanthanum chloride, also used more selective calcium channel blockers that inhibit members of individual calcium channel families.

Response: We thank the reviewer for highlighting the importance of examining more selective calcium channel blockers. In the revised manuscript, we tested a panel of Ca²⁺ signaling antagonists commonly used in plant research (reviewed by De Vriese et al. 2018. doi:10.3390/ijms19051506). Interestingly, our results show that the inhibitors of P-type Ca²⁺-ATPase specifically blocked the patchy induction pattern of *pICS1::NLS-3 × mVENUS* following *B. hordei* infection. These findings are highly informative, not only confirming the crucial role of Ca²⁺ signalling in the induction of *ICS1* expression, but also providing novel insights into how

coordinated intracellular and apoplastic Ca^{2+} homeostasis may contribute to immune-related Ca^{2+} signaling in plants. In the revised manuscript, we state:

*“We next examined the effects of additional Ca^{2+} signaling antagonists commonly used in plants. Interestingly, treatments with erythrosin B or eosin Y, antagonists of P-type Ca^{2+} -ATPase that mediate Ca^{2+} efflux, similarly suppressed the patchy induction pattern of $p\text{ICS1}::\text{NLS-3}\times m\text{VENUS}$ at 24 hpi following *B. hordei* infection (Supplemental Fig. 4). These results indicate that perturbation of Ca^{2+} fluxes disrupt pathogen-triggered ICSI expression, underscoring the essential role of Ca^{2+} homeostasis and signaling in this process.”* (lines 168-174).

In the Discussion section, we state:

*“Intriguingly, among the various Ca^{2+} signaling antagonists commonly used in both animal and plant systems, only eosin Y and erythrosin B, two potent inhibitors of ACA- P_{2B} type Ca^{2+} -ATPases, specifically suppress the infection-induced patchy expression patterns of ICSI and EDS5 (Supplemental Fig. 4). Members of the autoinhibited Ca^{2+} -ATPase (ACA) family belong to the P-type Ca^{2+} -ATPases, a class of active transporters that utilize ATP to pump Ca^{2+} out of the cytosol into intracellular compartments or the apoplast, thereby restoring basal cytosolic Ca^{2+} levels after transient Ca^{2+} elevations. This resetting activity is essential for maintaining Ca^{2+} homeostasis and shaping the amplitude, duration, and spatial distribution of Ca^{2+} oscillations. In *Arabidopsis*, ACA family members mediate active Ca^{2+} efflux into the apoplast, ER, and vacuole, thereby sustaining calcium gradients across subcellular compartments. Several recent studies have shown that genetic disruption of these pathways leads to altered immune and stress responses. Thus, ACA-type Ca^{2+} -ATPases act as key “reset” components of the calcium signaling network, maintaining homeostasis while enabling precise control over the signaling dynamics after exposure to specific stimuli. Our findings further suggest that ACA activity may contribute to infection-triggered SA biosynthesis and signaling through intercellular or cell to cell communication, although this mechanism warrants further genetic and molecular investigation.”* (lines 440-456)

13. The Discussion omits important points. How do the authors rationalize their findings in the context of a general SA-responsiveness of guard cells (e.g., <https://pmc.ncbi.nlm.nih.gov/articles/PMC6130018/>, <https://pmc.ncbi.nlm.nih.gov/article/s/PMC6512940/>, and many other publications)? Further, findings of guard cells in the context of immunity against filamentous pathogens are not properly integrated into the Discussion (e.g., <https://www.pnas.org/doi/10.1073/pnas.1922319117> and <https://link.springer.com/content/pdf/10.1186/s12870-024-05426-5.pdf>). It is also known that subsidiary cells of the stomatal complex can exhibit aberrant immunity to powdery mildew fungi. In barley *mlo* mutants, which are otherwise highly resistant against the barley powdery mildew pathogen, subsidiary cells remain susceptible (see https://www.apsnet.org/publications/phytopathology/backissues/Documents/1977Articles/Phyto67n05_678.PDF). The discrepancy between powdery mildew-resistant guard cells and powdery mildew-susceptible subsidiary cells would be worthwhile to discuss. A special role of subsidiary cells is also highlighted in the following work: <https://www.biorxiv.org/content/10.1101/2022.07.15.499463v1>.

Response: We appreciate the reviewer's expertise and broad knowledge in the field of stomatal-based immunity. We have carefully reviewed all of the literature mentioned by the reviewer and incorporated these studies into a more comprehensive discussion in the Discussion section. In the revised manuscript, we also present our rationales within the context of the general SA-responsiveness of guard cells.

“Numerous studies investigating ‘stomatal-based immunity’ across diverse plant species have highlighted stomatal closure as the first line of defense against bacterial and filamentous pathogens. Key signaling molecules, including abscisic acid (ABA), ROS, Ca²⁺, salicylic acid (SA) and jasmonic acid (JA), play crucial role in regulating the stomatal aperture in response to pathogen challenge or abiotic stress. With respect to SA involvement in regulating stomatal immunity, the existing literature primarily examines SA function acting at the tissue or whole-organ level in modulating GC behavior, rather than exploring the intrinsic capacity of GS themselves to synthesize or respond to SA”. (lines 470-477)

*“Together, these findings reveal for the first time that SA biosynthesis and signaling are not components of GC-autonomous immunity, immune responses intrinsic to individual GCs and distinct from their role in regulating stomatal movement. This cell-type-specific immune program in GCs versus pavement cells represents a conserved pattern across infections by fungal (*C. higginsianum*) and bacterial (*P. syringae*) pathogens (Fig. 4, Supplemental Fig. 9). Single-cell RNA-seq data further corroborate this distinction, demonstrating that GCs and pavement cells possess unique transcriptional signature under both fungal and bacterial pathogen challenge (Fig. 4, Supplemental Fig. 10). Collectively, these observations suggest that although GCs can perceive pathogen attack, their intrinsic and cell-autonomous immune responses are selective and mechanistically distinct from those of pavement cells.”* (lines 484-494)

Additionally, we have also discussed the discrepancy between powdery mildew-resistant guard cells and powdery mildew-susceptible subsidiary cells in the revised discussion section.

*“In monocots, each stoma comprises GCs accompanied by specialized subsidiary cells that support GC function. In barley mlo mutants, epidermal pavement cells exhibit strong resistance to barley powdery mildew, yet the subsidiary cells remain comparatively susceptible. Moreover, epidermal cells adjacent to invaded subsidiary cells also exhibit elevated susceptibility. These observations indicate that distinct epidermal cell types in barley differ in their vulnerability to powdery mildew and that successfully colonized subsidiary cells can influence neighboring cells, rendering them more susceptible to pathogen invasion. Unlike the powdery mildew fungi, *Zymoseptoria tritici*, the causal agent of *Septoria tritici* blotch in wheat, cannot directly penetrate the cuticle and instead enters the plant through stomata. Wheat cultivars with different types of resistance have been shown to block invasion by avirulent *Z. tritici* isolates during stomatal penetration. Using both wheat-*Z. tritici* and wheat-barley powdery mildew (*B. hordei*) pathosystems, Valente et al. (2022) demonstrated that subsidiary cells in the wheat epidermis were the first to respond to *Z. tritici* infection by forming cell wall papillae enriched with callose at fungus contact sites. Similarly, *B. hordei* infection triggers papilla formation and callose deposition in subsidiary cells, although the callose deposition patterns differ between two different fungal pathogens. Notably, no significant callose deposition is detectable in GCs. Furthermore, none of these studies assessed the immune response or immune status of the GCs themselves. Using*

Arabidopsis-powdery mildew pathosystems, our results are consistent with these observations, revealing a marked deficiency in callose deposition in GCs upon powdery mildew infection (Fig. 5A-B, Fig. 6A-C, Supplemental Fig. 13). Our systemic and comparative analyses of immune responses in GCs versus pavement cells will therefore be instrumental for advancing our understanding of epidermal cell-type-specific immunity. Nonetheless, stomatal architecture and mechanics vary substantially among plant species, a potential divergence in cell-type-specific immune mechanisms warrants further investigation.” (lines 526-549)

Reviewer #2 (Remarks to the Author)

Title: Deciphering Guard Cell-Autonomous Immunity Against Pathogenic Fungi (NCOMMS-25-57522)

1. *In this study, authors used transgenic immune reporter systems and live imaging to analyze cell-type specific immune responses in Arabidopsis leaves infected with different non-adapted and adapted pathogenic fungi. Study showed that while neighboring epidermal cells activated salicylic acid signaling and callose deposition, guard cells showed distinct immunity characterized by strong calcium influx, reactive oxygen species bursts, and hypersensitive cell death. Although the findings of the study are indeed novel and will advance our understanding of the autonomous and unique defense mechanisms occurring in guard cells, the actual mechanism of GC-mediated defense response is not clear even though the authors showed Ca signaling and ROS are involved. In addition, the authors need to address the following issues regarding some technical aspects and the claims made.*

Response: We thank the reviewer for their positive assessment of the novelty of our work and for their constructive comments. We agree that our study reveals previously unrecognized intrinsic, guard cell-autonomous immune features, and we appreciate the reviewer’s request for additional clarification regarding the mechanisms underlying guard cell-mediated defense. Our primary objective in this study was to map cell-type-specific immune outputs at high spatial and temporal resolution, rather than to fully delineate the signaling cascade involving Ca²⁺, ROS, and downstream components in guard cells. We concur that the mechanistic link between these signaling events and the observed hypersensitive responses warrants further exploration. To address this concern, we have performed several key experiments and analyses to elucidate potential mechanisms underlying guard cell-mediated defense. We briefly highlight these findings as follows:

First, our new imaging data demonstrate that guard cells do not activate SA biosynthesis or canonical SA signaling, even when they are directly infected. Guard cells adjacent to penetration sites show no induction of pICS1::NLS-3×mVENUS, pEDS5::NLS-mSCARLET, or SA-responsive *PR*-genes, whereas neighboring pavement cells strongly activate these pathways (Fig. 3A-D; Supplementary Fig. 8). This establishes that SA is not an intrinsic component of guard cell-autonomous immunity.

Second, we broadened our analysis to include additional fungal and bacterial pathosystems for both reporter assays and for single-cell RNA-seq. These data reveal that guard cells and pavement cells exhibit distinct pathogen-induced transcriptional signatures (Fig. 4; Supplementary Fig. 9-10). The transcriptomic profiles further support a mechanistic divergence in immune programming between the two epidermal cell types: pavement cells show strong engagement, and enrichment for SA biosynthesis and SA-responsive pathways, whereas guard cells are largely impaired in activating these responses.

Third, additional pharmacological perturbations using a panel of selective Ca²⁺ channel inhibitors further strengthen the causal link between Ca²⁺ signaling and infection-triggered SA biosynthesis. Our results show that dynamic Ca²⁺ influx, rather than elevated intracellular Ca²⁺ concentration per se, is required for pathogen-induced SA biosynthesis (Supplementary Fig. 4).

Fourth, our new data show that guard cells mount a faster and more amplified Ca²⁺ influx and ROS burst upon pathogen perception than pavement cells, and that these early signaling events precede SA biosynthesis in pavement cells (Fig. 5D-G; Supplementary Fig. 11). Notably, similarly robust ROS bursts occur during infection by the adapted powdery mildew *E. cichoracearum*, indicating that guard cells elicit stronger and more rapid immune responses than pavement cells upon pathogen detection.

Fifth, we provide new genetic evidence that hypersensitive cell death in guard cells occurs independently of SA accumulation. Using an epidermis-specific SA-depleted line (*pAtML1::NahG-mSCARLET*), we show that *B. hordei*-induced guard cell death is unaffected by the loss of SA (Supplementary Fig. 12C-E). These findings suggest that infection-triggered guard cell death is likely driven by Ca²⁺-dependent ROS signaling rather than activation of the SA pathway.

Finally, we substantially revised the Discussion to clarify how guard cells deploy a distinct, intrinsic, cell-autonomous immune program that is mechanistically separate from both pavement-cell immunity and the classical stomatal aperture-based immune responses.

Together, these new experiments clearly establish a clear mechanistic framework for guard cell-autonomous immunity, centered on strong Ca²⁺ influx, amplified ROS production, and SA-independent cell death, thereby fully addressing the reviewer's concern.

2• How many transgenic lines exhibited similar expression patterns for pICS1::ICS1-mSCARLET and pEDS5::NLS-mSCARLET? Since these lines have not been reported before, please present data from at least 2–3 independent lines showing consistent expression patterns in response to fungal pathogens. These data can be included as supplementary material.

Response: We generated at least 5 independent transgenic lines for both *pICS1::ICS1-mSCARLET* and *pEDS5::NLS-mSCARLET* constructs, all of which exhibited similar expression patterns. In the revised manuscript, we have included an additional independent line for each reporter (See Supplemental Fig. 2B, Supplemental Fig. 3A).

3• This study used several mutants (e.g., eds5, sard1, cbp60g, camta123, etc.) related to different defense-related pathways, which were crossed with reporter lines. Although the primer

sequences used for genotyping are provided in the supplementary material, no data confirming the genotyping of these lines for homozygosity and knockout status of the target genes are included. Please provide these data as supplementary material.

Response: As suggested by the reviewer, we have included the genotyping results confirming the homozygosity and knockout status of the target genes for these lines in the revised manuscript (please see Supplemental Fig. 5A-B, Supplemental Fig. 6B, Supplemental Fig. 7B).

4• Although the study analyzed different fungi, terms such as 'pathogen attack,' 'pathogen-induced SA,' or 'pathogen-induced' such as in lines 53 and 57, are used throughout the manuscript when reporting observations in response to fungal infection. Please be specific to fungal pathogens, as responses to other pathogens (bacterial or viral) were not tested and may differ.

Response: In this revised manuscript, we have extended our study to include additional fungal (*Colletotrichum higginsianum*) and bacterial (*Pseudomonas syringae*) pathogens. In addition, we have revised the terminology throughout the text to specify the type of pathogen accordingly.

5• How would you explain a faint cyan color in the neighboring cells for adapted fungus (supplemental Fig. 1A)?

Response: The ICS1 reporter transgene pICS1::NLS-3×mVENUS exhibits basal expression under normal conditions, as shown in Fig. 1A-B. In the revised manuscript, we carefully examined the interaction between the adapted powdery mildew pathogen *E. cichoracearum* and Arabidopsis. Following *E. cichoracearum* infection at 26 hpi, we observed a patchy induction pattern of NLS-3×mVENUS in the ICS1 reporter line, resembling the pattern observed during *B. hordei* infection. The revised manuscript has been updated to reflect these findings.

6• In Fig. 2D, the pICS1-driven reporter construct shows low activity; therefore, the claim in lines 187–188 that NLS-3×mVENUS signals are completely absent in the *cbp60g sard1* mutants does not appear to be accurate. Additionally, there is no evident saturation of pICS1::NLS-3×mVENUS signals in the *camta1/2/3* mutant upon *B.g. hordei* infection. In fact, the signal appears weaker in *B.g. hordei*-infected tissue (Fig. 2D) compared to the non-infected control (Fig. 2C). Why is signal quantification provided for the reporter in non-infected tissues in Supplemental Fig. 4B (based on Fig. 2C), but not for the infected tissues shown in Supplemental Fig. 4 (based on Fig. 2D)?

Response: We meant the patchy induction pattern of NLS-3×mVENUS signals was absent in the *cbp60g sard1-2* mutant background. To avoid any confusion, we have revised our statement.

“Upon *B. hordei* infection at 24 hpi, NLS-3×mVENUS signals were strongly induced at the infection sites in WT plants but remained at basal levels in the *cbp60g sard1-2* mutants.” (lines 196-198)

To address the reviewer's concern regarding signal intensities, we have included quantitative measurements for the infected tissues (see Supplemental Fig. 5D).

7• Line 217-188, the authors mention examining the reporter gene expression in infected GCs. How often do they find this? Some numbers or frequency will help.

Response: We observed at least 10 independent sites where guard cells were directly infected by *B. hordei*. None of these infection events triggered ICS1 expression. The corresponding data have been included in the revised figure legend of Fig. 3D.

8• Lines 267–268: There appears to be callose deposition in one of the guard cells (GCs) of the kidney-shaped structure in Fig. 4B and C. Although it seems slightly away from the penetration site, it still appears to be part of the GC. A similar pattern of callose deposition is observed in the GCs in Supplemental Fig. 7B for the adapted fungal pathogen. Therefore, it is unclear from these images whether the callose was deposited by the GCs or by neighboring pavement cells, as callose is always deposited in the apoplast (outside the cell membrane).

Response: We appreciate the reviewer's careful attention to the callose deposition patterns. In our study, callose was visualized using aniline blue staining in a basic solution (pH 9.5). Because aniline-blue-stained callose produces strong autofluorescence, the resulting fluorescence signal can extend into neighboring cells, potentially giving the impression of signal within guard cells. The apparent signal inside portions of the guard cell is therefore most likely due to the fluorescence saturation caused by the intense aniline blue staining.

In Fig. 4B&C (now revised as Fig. 5B&C) and Supplemental Fig. 7B (revised Supplemental Fig. 13B), the callose signals originate from the surrounding pavement cells rather than from the guard cells. In both Fig. 5B&C and Supplemental Fig. 13B, the discontinuous pattern of callose deposition, including the distinct gap corresponding to the intercellular space between two pavement cells, clearly indicates that the deposition occurs in pavement cells adjacent to the infected guard cell. Supplemental Videos 2 and 3 support this conclusion by providing three-dimensional visualization of the signal distribution.

Additionally, we note that callose deposition is absent at the penetration sites of guard cells themselves, both for the non-adapted *B. hordei* (Fig. 5B & C) and the adapted *E. cruciferarum* (Supplemental Fig. 13B), further indicating that guard cells do not deposit callose in these instances.

9• The GC-mediated defense against adapted pathogen is intriguing but the mechanism is not shown. For non-adapted pathogen, the authors show Ca signaling ROS play a role in cell defense. Is it the same for adapted pathogens? This can be easily tested. This observation also warrants more discussion. Why an adapted pathogen not able to suppress defense response in GCs? Does GCs lack components needed for fungal cell differentiation and development?

Response: We thank the reviewer for their insightful comments. In response to the suggestion, we conducted additional experiments to examine whether Ca²⁺ influx and ROS accumulation also

occur during infection by the host-adapted powdery mildew. These new data (Supplemental Fig. 11C-H) now allow a direct comparison of guard cell responses to both non-adapted and adapted pathogens.

Our results show that infection by the adapted powdery mildew *E. cichoracearum* triggers pronounced Ca^{2+} elevations and strong ROS accumulation in guard cells, surpassing the responses observed in adjacent pavement cells (Supplemental Fig. 11C-F). As now described in the revised text:

“Notably, infection by compatible E. cichoracearum induces higher Ca^{2+} elevation and ROS accumulation in GCs adjacent to the infected pavement cell than in the surrounding pavement cells at 22-26 hpi (Supplemental Fig. 11C-F). These results suggest that GCs respond more vigorously to pathogen challenges than pavement cells, demonstrated by their more rapid and pronounced Ca^{2+} elevation as well as heightened intracellular ROS accumulation.” (Lines 359-364).

Thus, guard cells mount enhanced early Ca^{2+} - and ROS-associated responses to both adapted and non-adapted pathogens, and their signaling amplitude remains high even during compatible interactions.

In response to the reviewer’s second point regarding potential mechanisms underlying GC-mediated defense against adapted pathogens, we refer to our detailed reply to Comment 1. Although guard cells are impaired in SA biosynthesis and SA signaling, our new results show that they exhibit a faster and more amplified Ca^{2+} influx and ROS burst than pavement cells upon perception of the non-adapted *B. hordei* infection (Supplemental Fig. 11C-E). Moreover, stronger ROS accumulation in guard cells relative to pavement cells is also observed in regions adjacent to infection sites of the adapted powdery mildew *E. cichoracearum* (Supplemental Fig. 11F-H). We further demonstrate that *B. hordei*-induced cell death accompanied by ROS accumulation occurs despite the absence of SA accumulation (Supplemental Fig. 12A-E).

Taken together, these findings support our proposal that early immune activation, characterized by Ca^{2+} influx and ROS burst, is robustly executed in guard cells, potentially contributing to their incompatibility even during otherwise compatible interactions. Nonetheless, we cannot exclude the possibility that guard cells may simply lack components required for fungal differentiation and development, as the reviewer suggested. We now discuss potential mechanisms underlying guard cell innate immunity more in great depth in the revised Discussion section.

10• Discussion section is very poorly written. It is just a summary of results and repetitive rather than discussion with reference to previous work. There are numerous studies regarding the role of GCs in plant defense. Surprisingly, only three additional references are added in the discussion! In addition, the authors could hypothesize or come up with a model regarding how the GC-mediated defense response is triggered? For example, is the defense response triggered directly by PAMPs or internal signal from neighboring cells?

Response: We thank the reviewer for this important comment. In response, we have substantially rewritten, expanded, and reorganized the Discussion to provide a more integrative and mechanistic

interpretation of our findings, firmly grounded in the existing literature. The revised Discussion now includes the following improvements:

- a) A deeper integration of our live-cell imaging data using immune reporters with prior studies on spatial single-cell immune transcriptomics (lines 415-432; 487-494).
- b) A new conceptual framework explaining how dynamic Ca^{2+} homeostasis regulates SA biosynthesis through intercellular communication (lines 433-456).
- c) An expanded comparison between classical stomatal-based immunity, where stomata serve as active immune gates restricting pathogen entry, and our findings on intrinsic, guard cell-autonomous immunity (lines 470-494).
- d) Additional discussion of potential mechanisms underlying guard cell immunity, including Ca^{2+} /ROS signaling, CERK1-mediated chitin perception (lines 500-517), and Ca^{2+} /ROS-mediated cell death independent of SA signaling (lines 517-525).
- e) A proposed rationale for the incompatibility of guard cells with adapted pathogens (lines 517-525).
- f) A comprehensive comparison of stomatal cell-type-specific responses across wheat-rust, wheat-*Z. tritici*, and barley-*B. hordei* pathosystems, drawing on the studies highlighted by Reviewer 1 (lines 526-549).

We believe these major revisions substantially strengthen the Discussion and fully address the reviewer's concerns.

Minor Comments

11• *Figure legend 4A. is it really a "cyan color or magenta" representing aniline blue staining?*

Response: We have double checked the figure legend and corrected the description in the revised version of figures.

12• *Also include a paragraph on comparing cell-type specific defense responses (Ca^{2+} signaling, ROS accumulation, SA production) between fungal and bacterial pathogens in the discussion section.*

Response: we have added a paragraph to discuss cell-type specific defense responses in the revised manuscript. Please see lines 470-477 and 484-494.

"Numerous studies investigating 'stomatal-based immunity' across diverse plant species have highlighted stomatal closure as the first line of defense against bacterial and filamentous pathogens. Key signaling molecules, including abscisic acid (ABA), ROS, Ca^{2+} , salicylic acid (SA) and jasmonic acid (JA), play crucial role in regulating the stomatal aperture in response to pathogen challenge or abiotic stress. With respect to SA involvement in regulating stomatal immunity, the existing literature primarily examines SA function acting at the tissue or whole-organ level in modulating GC behavior, rather than exploring the intrinsic capacity of GS themselves to synthesize or respond to SA"

*“Together, these findings reveal for the first time that SA biosynthesis and signaling are not components of GC-autonomous immunity, immune responses intrinsic to individual GCs and distinct from their role in regulating stomatal movement. This cell-type-specific immune program in GCs versus pavement cells represents a conserved pattern across infections by fungal (*C. higginsianum*) and bacterial (*P. syringae*) pathogens (Fig. 4, Supplemental Fig. 9). Single-cell RNA-seq data further corroborate this distinction, demonstrating that GCs and pavement cells possess unique transcriptional signature under both fungal and bacterial pathogen challenge (Fig. 4, Supplemental Fig. 10). Collectively, these observations suggest that although GCs can perceive pathogen attack, their intrinsic and cell-autonomous immune responses are selective and mechanistically distinct from those of pavement cells.”*

13• Line 108: “emerge: instead of “merge”?”

Response: We have revised the statement in the revised manuscript.

“By 21 hpi, however, visible induction of nuclear-associated fluorescent signals became apparent in “Patient 0” and adjacent L1 cells (Fig. 1A-B).” (lines 115-117)

14• Line 213: replace GSs (GCs)

Response: Corrected.

15• Line 282: Expand or explain H2DCF-DA. Readers may not be familiar with this.

Response: We have provided additional explanation of H2DCF-DA, see the revised manuscript.

“H2DCF-DA, a cell-permeable fluorescent probe that is deacetylated by intracellular esterases and oxidized by ROS, enabling visualization of ROS accumulation in live GCs.” (lines 350-351)

16• Line 294: replace “non-adapted fungus pathogens” with “non-adapted fungal pathogens”.

Response: Corrected.

17• Throughout the manuscript either use *A. thaliana* or *Arabidopsis* (common name with no italics).

Response: We have revised it to “*Arabidopsis*” throughout the revised manuscript.

18• The title of the manuscript can be a bit more descriptive and interesting to describe the actual findings of the manuscript.

Response: In this revised manuscript, we have updated the title to “*Cell-type-specific immune programs orchestrate spatial defense in the Arabidopsis leaf epidermis*”.

Reviewer #3 (Remarks to the Author):

Response: We thank this reviewer for their time and effort to provide constructive comments. Please see our point-by-point responses above.

Response to Reviewers' Comments

We sincerely thank the reviewers for the time and effort they devoted to evaluating our manuscript. We greatly appreciate their thoughtful and supportive comments.

Reviewers' comments:

Reviewer #1 (Remarks to the Author):

The resubmitted version of the manuscript by Song and co-workers shows impressive improvements. The authors addressed all my concerns and now provide several new experimental data and in silico analyses of single-cell RNA-Seq data. Major improvements cover the following aspects:

- 1.) Marker fluorescence intensities were quantified. This approach further strengthens the claims of the authors.*
- 2.) Pre-existing single cell RNA-Seq data from infection experiments (conducted by other groups) with *C. higginsianum* and *P. syringae* were analyzed. Careful examination of these data sets supports the key conclusions of the present manuscript. Using pre-existing published single-cell transcriptomic data is a creative approach to address this issue. Supplementing this, infection experiments with ICSI reporter lines were performed to examine for a putative cell type-specific activation of the SA signaling pathway in these interactions.*
- 3.) In addition to the NahG transgene, the sid2 mutant was used for crossings with the ICSI reporter line. The results of both approaches are consistent. I do appreciate the rationale of the authors to keep the NahG transgene, as this shows a stronger effect.*
- 4.) A panel of more specific calcium inhibitors has been tested (at single concentrations only, though), which revealed a key role for P-type calcium ATPases in the activation of pathogen-induced ICSI expression. Maybe a short note could be added to justify the concentrations used.*
- 5.) The Discussion was largely expanded and improved. It covers many more aspects, placing the findings in a broader context. It now has sufficient breadth and depth.*

In addition to the key issues, the authors also dealt with the minor points mentioned. I have the impression that also the concerns of reviewer #2 were addressed adequately to the most part, as we shared some similar comments, e.g., regarding the lack of more mechanistic insights.

In summary, I feel that the current version has improved substantially. The authors have made a major and serious effort to advance their study. In my opinion, the work has reached the level required for publication in a journal like Nature Communications. Unraveling cell type specificity in immune responses is a topic of broad scientific relevance. I congratulate the authors on an important advancement in our understanding of plant immunity.

Reviewer #2 (Remarks to the Author):

The authors have substantially improved the manuscript by adding a new dataset that supports their findings on distinct defense responses in pavement and guard cells, namely salicylic acid signalling and callose deposition in pavement cells and strong Ca^{2+} influx, amplified ROS production, and SA-independent cell death in guard cells. Furthermore, the conservation of these cell-type-specific responses to both bacterial and fungal infections make the study particularly interesting and novel. Additionally, the authors have successfully addressed the issues raised in the previous submission

Reviewer #3 (Remarks to the Author):
